# The Influence of Graphene Oxide-Fe_3_O_4_ Differently Conjugated with 10-Hydroxycampthotecin and a Rotating Magnetic Field on Adenocarcinoma Cells

**DOI:** 10.3390/ijms25020930

**Published:** 2024-01-11

**Authors:** Magdalena Jedrzejczak-Silicka, Karolina Szymańska, Ewa Mijowska, Rafał Rakoczy

**Affiliations:** 1Laboratory of Cytogenetics, West Pomeranian University of Technology in Szczecin, Klemensa Janickiego 29, 71-270 Szczecin, Poland; 2Department of Physicochemistry of Nanomaterials, Faculty of Chemical Technology and Engineering, West Pomeranian University of Technology in Szczecin, Piastow Ave. 42, 71-065 Szczecin, Poland; karolina.szymanska@zut.edu.pl (K.S.); emjiowska@zut.edu.pl (E.M.); 3Institute of Chemical Engineering and Environmental Protection Process, Faculty of Chemical Technology and Engineering, West Pomeranian University of Technology in Szczecin, Piastow Avenue 42, 71-065 Szczecin, Poland

**Keywords:** graphene oxide-Fe_3_O_4_ nanocomposite, rotating magnetic field (RMF), MCF-7 cell culture, cell viability, biomedical application, drug delivery strategies

## Abstract

Nanoparticles (e.g., graphene oxide, graphene oxide-Fe_3_O_4_ nanocomposite or hexagonal boron nitride) loaded with anti-cancer drugs and targeted at cancerous cells allowed researchers to determine the most effective in vitro conditions for anticancer treatment. For this reason, the main propose of the present study was to determine the effect of graphene oxide (GO) with iron oxide (Fe_3_O_4_) nanoparticles (GO-Fe_3_O_4_) covalently (c-GO-Fe_3_O_4_-HCPT) and non-covalently (nc-GO-Fe_3_O_4_-HCPT) conjugated with hydroxycamptothecin (HCPT) in the presence of a rotating magnetic field (RMF) on relative cell viability using the MCF-7 breast cancer cell line. The obtained GO-Fe_3_O_4_ nanocomposites demonstrated the uniform coverage of the graphene flakes with the nanospheres, with the thickness of the flakes estimated as ca. 1.2 nm. The XRD pattern of GO–Fe_3_O_4_ indicates that the crystal structure of the magnetite remained stable during the functionalization with HCPT that was confirmed with FTIR spectra. After 24 h, approx. 49% and 34% of the anti-cancer drug was released from nc-GO-Fe_3_O_4_-HCPT and c-GO-Fe_3_O_4_-HCPT, respectively. The stronger bonds in the c-GO-Fe_3_O_4_-HCPT resulted in a slower release of a smaller drug amount from the nanocomposite. The combined impact of the novel nanocomposites and a rotating magnetic field on MCF-7 cells was revealed and the efficiency of this novel approach has been confirmed. However, MCF-7 cells were more significantly affected by nc-GO-Fe_3_O_4_-HCPT. In the present study, it was found that the concentration of nc-GO-Fe_3_O_4_-HCPT and a RMF has the highest statistically significant influence on MCF-7 cell viability. The obtained novel nanocomposites and rotating magnetic field were found to affect the MCF-7 cells in a dose-dependent manner. The presented results may have potential clinical applications, but still, more in-depth analyses need to be performed.

## 1. Introduction

According to the Global Burden of Diseases, Injuries, and Risk Factors Study 2019 (GBD 2019) [1], the problem of cancer is more topical than ever and is responsible for one in six deaths worldwide [2]. The GBD 2019 showed that the rates of cancer incidence have increased from 14.1 million and 8.2 million cancer-related deaths in 2012 to 23.6 million new cancer cases and 10.0 million cancer deaths globally. Moreover, the GBD 2019 indicates an estimated 250 million (235–264 million) disability-adjusted life years (DALYs) due to cancer [1].

An adequate diagnosis is a key factor in cancer therapy, but an effective cancer treatment depends on the type of cancer (e.g., leukemia, lymphoma, breast cancer, cervical cancer, lung cancer, testicular seminoma). The treatment of specific cancers is a highly complex process, which may involve traditional treatment such as surgery and/or chemotherapy, and radiotherapy [2]. It is also known that each modality can cause different side-effects. Therefore, many studies have recently focused on alternative forms of therapies that may give possibilities of treating degenerative diseases and improve the treatment efficiency of diseases such as cancer [3,4,5], including, e.g., stem cell therapy, targeted therapy, ablation therapy, nanoparticles, chemodynamic therapy, sonodynamic therapy, and ferroptosis-based therapy [3]. One promising approach to cancer therapy can be the magnetic localized hyperthermia induced by a magnetic field and magnetic nanomaterials combined with an anti-tumor drug injected into the tumor [6]. The benefits of the magnetically triggered formulation in cancer treatment are: (i) a reduction of the amount of a drug required to achieve an effective therapeutic dose, (ii) targeted and controlled drug delivery and effective induction of apoptosis in tumor cells, without affecting healthy tissues surrounding the tumor [4]. These effects can be obtained due to the irregular and poor blood flow in the tumor that causes slow heat dissipation, which in turn makes cancer cells more thermosensitive than normal cells [6].

Therapeutic heating can be achieved using particles with superparamagnetic properties (e.g., a nanocomposite of graphene oxide nanosheets conjugated with magnetic Fe_3_O_4_ nanoparticles or superparamagnetic iron oxide nanoparticles [SPIONs]) [4,7] and an external alternating current (AC) magnetic field [8]. Magnetic particles release heat in the applied field by the rotation of the magnetization within the particles—Néel relaxation—or by the rotation of the particles against the dispersed medium—Brownian relaxation [4]. The nanoparticles targeted at specific cancer cells exposed to a time-varying magnetic field will increase to a temperature (above 45 °C to induce cell death) determined by the magnetic properties of the NPs and the strength of the external alternating current magnetic field. Exposure to the rotating magnetic field (RMF) creates hyperthermia, but also a nano-mixing process, which affects cancer cells, resulting in an increase of membrane permeability via magneto-cell-lysis and magneto-cell-poration [9,10]. The determination of the magnetic behavior of the nanoparticles, including their superparamagnetic nature and coercivity is crucial for, among others, optimizing the synthesis of the magnetite [11]. These types of information—the characterization of iron oxide nanoparticles in biological samples (cells and tissues)—can be obtained by using SQUID (superconducting quantum interference device) magnetometry [11]. Moreover, detection of ferromagnetic materials in different tissues (their orientation toward external magnetic fields) can be detected by vibrating sample magnetometry (VSM). Whereas the magnetization process and the subsequent response of the atomic or electronic spins according to magnetic field can be recorded by magneto-optic Kerr effect (MOKE) and magnetic resonance techniques (like nuclear magnetic resonance—NMR, electron paramagnetic resonance—EPR, ferromagnetic resonance—FMR) [12].

Current studies show great interest in hyperthermia in cancer treatment. Liu et al. analyzed the effect of the magnetoporation of the MCF-7 cell line exposed to a 40 mT RMF and multiwall carbon nanotubes (MWCNT) [9]. Bai et al. tested the superparamagnetic graphene oxide nanosheet–Fe_3_O_4_ nanoparticle (GO–Fe_3_O_4_) hybrids in an AC magnetic field and found that this nanocomposite could be a useful material for the localized hyperthermia in cancer treatment [5]. Chen et al. presented a hybrid GO with magnetite nanoparticles loaded with doxorubicin hydrochloride (DOX)—an anti-cancer drug and considered the described hybrid as a potential candidate for controlled drug delivery. The use of anticancer drugs, such as paclitaxel (PTX), doxorubicin (DOX), hydroxycamptothecin (HCPT), docetaxel, 5-fluorouracil (5-FU), podophyllotoxin (PTOX) that can be controlled released by an external magnetic field-induced biodistribution system gave important information about the magnetic hyperthermia-based treatment, drug release kinetics, and targeting ability and improves the efficiency of combination therapy [4,8,13,14]. The graphene oxide nanoflakes were also successfully covalently conjugated with the anticancer drug methotrexate (MTX) and the anticancer activity of that functionalized nanomaterial (MTX-GO) against the MCF-7 adenocarcinoma cell line was confirmed [15]. Another study shows that graphene oxide nanosheets exhibit high DOX loading capacity—2.35 mg of DOX per 1.0 mg of GO [16,17]. Graphene oxide loaded with adriamycin (ADR) can also exhibit a reversal effect in the MCF-7/ADR-resistant breast cancer cell line. The results obtained by Wu et al. showed that ADR-GO tended to accumulate in MCF-7/ADR and had a higher cytotoxicity than free adriamycin [15]. This result suggests that the nanocomposite of the combined GO and ADR effectively reverses cancer cells’ ADR-resistance [15]. It was also found by Liu et al. that modifications of GO can effectively improve its stability and biocompatibility by using branched, biocompatible polyethene glycol (PEG) [18]. The PEGylated graphene oxide (GO-PEG) conjugated with camptothecin analogue—7-ethyl-10-hydroxycamptothecin (SN38) exhibits high aqueous solubility and, at the same time, this modification retains the high potency of free SN38 [18]. Other research groups designed nanomaterials based on GO and hematin-terminated dextran (HDex). The nanohybrid materials obtained in that way can be used as a drug delivery platforms, e.g., GO-HDex loaded with DOX with a high loading capacity—3.4 mg/mg GO effectively kills the cancer cells [19].

Due to recent studies that are focused on the development of safe and efficient cancer nanomedicines [3], the present work is aimed at developing an effective dual therapy using ferromagnetic nanoparticles with a rotating magnetic field (RMF) and anticancer drugs. Here, the human breast adenocarcinoma cell line, MCF-7, was exposed to the nanocomposite of a graphene oxide nanosheet with Fe_3_O_4_ nanoparticles (GO-Fe_3_O_4_), loaded with hydroxycamptothecin (covalently and non-covalently attached) and exposed to a rotating magnetic field (RMF) at different magnetic inductions (***B***) ranging from 1.23 to 10.06 mT, incubated for over 8 h. The combination of factors, such as nanocomposite loading, the type of drug, type of conjugation with the anti-tumor drug and the magnetic field allowed us to determine the most effective in vitro conditions for cancer treatment.

## 2. Results and Discussion

### 2.1. Materials Characterization

The sample morphology was examined using transmission electron microscopy (TEM). Representative TEM images of GO and Fe_3_O_4_ nanospheres are shown in Figure 1a,b, while pictures of the GO-Fe_3_O_4_ nanocomposite are presented in Figure 1c,d. The TEM image of GO (Figure 1a,b) demonstrates the shape of the flakes, while the Fe_3_O_4_ image (Figure 1c,d) spherical shape of iron oxide nanoparticles. Based on the TEM images of the Fe_3_O_4_ nanospheres, the diameter distribution of iron oxide has been calculated and presented in the graph in Figure 2d. In the next step, the spheres and GO were covalently functionalized to create the novel, hybrid material (functionalized with glycine that is easily adsorbed onto the iron oxide surface) (Figure 1c,d). The images demonstrate the uniform coverage of the graphene flakes with the nanospheres.

Atomic force microscopy (AFM) was employed to explore the morphology, size and the thickness of graphene oxide. The AFM image of GO inserted in Figure 2 confirmed the flake morphology of the material observed previously on TEM. The size distribution of graphene oxide flakes is presented on the graph in Figure 2c,d and it ranges from less than 100 nm to 4 µm. The thickness of the flakes can be estimated as ca. 1.2 nm based on the height profiles of graphene oxide flakes (Figure 2b).

The X-ray diffraction technique (XRD) is a rapid and useful technique for investigating the crystal structure of materials. Figure 3a shows the XRD pattern of GO, which is dominated by an intense and narrow peak at 2θ = 11.5° associated with the reflection in (002) planes of well-ordered graphene oxide layers. The presence of this peak is proof of the successful oxidation of graphite. The small peak at ca. 26° can be assigned to traces of the starting material (graphite flakes). Figure 3b presents the XRD spectrum of Fe_3_O_4_ nanospheres. The relative intensities of the peaks and their positions correspond to the standard XRD data for magnetite (ICSD 65339). The XRD pattern of GO–Fe_3_O_4_ indicates that the crystal structure of the magnetite remained stable during the functionalization (Figure 3c).

The efficiency of the functionalization of the GO-Fe_3_O_4_ with HCPT was confirmed with the FTIR spectra of both nanocomposites (HCPT covalently and non-covalently attached to the GO-Fe_3_O_4_) were measured. The peaks derived from the Fe-O bond are present in both spectra. Additional peaks were observed in the c-GO-Fe_3_O_4_-HCPT spectrum (two additional peaks originated from the covalent bonding between GO and magnetite) in comparison to the nc-GO-Fe_3_O_4_-HCPT nanocomposite spectrum, where the lack of chemical bonds between the nanoparticle carrier and the drug (HCPT) was observed. The c-GO-Fe_3_O_4_-HCPT spectrum is characterized by peaks at 1359 cm^−1^ derived from C-N bond and 3245 cm^−1^ arising from N-H (Figure 4). Therefore, the presence of the abovementioned peaks proved the propitious covalent functionalization of the GO-Fe_3_O_4_ carrier with the anti-cancer drug. The remaining minor peaks in the range of 613 cm^−1^ to 1405 cm^−1^ in both spectra are associated with the presence of hydroxycamptothecin [20].

### 2.2. The Stability of the Dispersions of nc-GO-Fe_3_O_4_-HCPT and c-GO-Fe_3_O_4_-HCPT Nanocomposites

The stability of the dispersions of nc-GO-Fe_3_O_4_-HCPT and c-GO-Fe_3_O_4_-HCPT in different concentrations over time can be estimated based on Figure 5a,b, respectively. Overall, the highest decrease in stability throughout the whole measurement was observed for the highest concentrations of the dispersions of both tested materials. Except for 100 µm mL^−1^, the dispersions remain practically stable up to a few hours, then the stability decreases gradually. It is also worth noting that the decrease in instability over time is smaller for the dispersions of the material with noncovalent bonding.

### 2.3. HCPT Loading and Release

The drug loading for nc-GO-Fe_3_O_4_ was estimated to be ~12.1%, while for c-GO-Fe_3_O_4_—~9.5%. The experiments on the release of HCPT incorporated in the prepared nanocomposites were conducted in PBS (pH 7.4) at 37 °C. The UV absorption spectra were measured in the supernatants to assess the amount of HCPT released at specified time intervals. The resulting data are presented in Figure 6. The release profiles showed a steady release pattern after the initial burst. After the 24 h, approx. 49% and 34% of the anti-cancer HCPT drug was released from nc-GO-Fe_3_O_4_-HCPT and c-GO-Fe_3_O_4_-HCPT, respectively. The stronger bonds in c-GO-Fe_3_O_4_-HCPT resulted in a slower release of a smaller drug amount from the nanocomposite in comparison to nc-GO-Fe_3_O_4_-HCPT during 130 h of the experiment. Similar observations were made in case of GO [21] and micelles bearing covalently and non-covalently attached drugs. The materials with covalent conjugation appear to be more stable than those with physically entrapped drugs [22]. Also, covalently coupled drug–dendrimer conjugates were stable in both water and PBS solution. They are claimed to be better suited for targeted drug delivery than drug–dendrimer systems based on noncovalent interactions, as they do not release the drug prematurely in the biological environment [23]. Further investigations such as drug release experiments in the presence of RMF will be performed in the future.

### 2.4. Morphology of MCF-7 Cells after Combined Exposure to Nanocomposites and Rotating Magnetic Field

The morphology of MCF-7 cells was monitored after 8 h of exposure to a rotating magnetic field (RMF). The morphology of the unexposed cells (control culture, Appendix A) was compared to the morphology of cells incubated with c-GO-Fe_3_O_4_-HCPT (Appendix A) and nc-GO-Fe_3_O_4_-HCPT (Appendix A). Cell cultures exposed to nanomaterials at the lowest concentration (3.125 µgmL^−1^ and 6.25 µgmL^−1^) exhibited similar morphology to the control culture. In both cases, in cultures treated with c-GO-Fe_3_O_4_-HCPT and nc-GO-Fe_3_O_4_-HCPT nanocomposites, the cells demonstrated a polygonal shape (the presence of single vacuoles was indicated by green arrows) and the used nanoparticles did not form aggregates (Appendix A). MCF-7 cells exposed to c-GO-Fe_3_O_4_-HCPT at a dose of 12.5 µgmL^−1^ presented a normal cellular shape and density (Appendix A), but in the cell culture incubated with nc-GO-Fe_3_O_4_-HCPT, single cells were floating in the culture medium (Appendix A; red arrows). Moreover, the nc-GO-Fe_3_O_4_-HCPT form small aggregates (outside the cells; Appendix A; blue arrows). When cells were exposed to a higher dose of c-GO-Fe_3_O_4_-HCPT (25.0 µgmL^−1^ and 50.0 µgmL^−1^), they again displayed a typical shape and the nanocomposite at these concentrations formed small aggregates (outside the cells; blue arrows) (Appendix A). At 25.0 and 50.0 µgmL^−1^ concentrations of nc-GO-Fe_3_O_4_-HCPT, the cells also had a normal shape, similar to the cells incubated with nc-GO-Fe_3_O_4_-HCPT, however, aggregates of nc-GO-Fe_3_O_4_-HCPT were observed in the cytoplasm (Appendix A; blue arrows). This nc-GO-Fe_3_O_4_-HCPT nanocomposite at the concentration of 50.0 µgmL^−1^ accumulated in the cytoplasm near the nuclei and formed brownish aggregates of different sizes (Appendix A; blue arrows).

### 2.5. The MCF-7 Cell Line Response to Graphene Oxide-Fe_3_O_4_ and Hydroxycamptothecin

Firstly, our study determined the effect of the GO-Fe_3_O_4_ hybrid on the metabolism of the human breast adenocarcinoma MCF-7 cell line. The influence of the GO-Fe_3_O_4_ hybrid nanoparticles was determined via WST-1, LDH leakage and neutral red assays. The relative cellular activity (the WST-1 assay results) of MCF-7 cells is presented in Figure 7a. It can be stated that the mitochondrial activity of chosen in vitro model was not affected by the hybrid nanoparticles in a dose-dependent manner. The data obtained from both assays showed a small cytotoxic effect on MCF-7 cells. The GO-Fe_3_O_4_ hybrid reduced mitochondrial activity the most at a concentration range from 25.0 to 50.0 µgmL^−1^. It was recorded that the relative viability of MCF-7 cells decreased to ~80% vs. control cell cultures. The LDH assay analysis showed that the effect of the GO-Fe_3_O_4_ hybrid was not dose-dependent (Figure 7a). The highest LDH leakage was observed at a dose of 12.5 µgmL^−1^ (an ~10% LDH leakage was noticed at the aforementioned concentration). A lower or similar cytotoxicity impact effect on LDH leakage was found for 3.125, 6.25, 25.0 and 50.0 µgmL^−1^ of the tested nanocomposites (~2–5%). Moreover, the neutral red assay performed in our study confirmed the low cytotoxic effect of GO-Fe_3_O_4_ on MCF-7 cells. Yang et al. prepared GO-Fe_3_O_4_ hybrid NPs loaded with doxorubicin hydrochloride (DXR) for controlled targeted drug delivery and release [24]. Yang et al. also produced graphene oxide-magnetic nanoparticle (GPO-Fe_3_O_4_ and GPO-do-Fe_3_O_4_) and characterized its magnetic and electrical properties for biomedical applications [25]. Chen et al. reported that GO-Fe_3_O_4_ might have a potential clinical application, and it did not affect cell viability and proliferation [26]. The potential clinical utility is strictly connected with the high biocompatibility of the tested nanomaterial [27]. In our previous study, it was found that the deposition of iron oxide nanoparticles on a graphene oxide platform (with different size distribution, broad size from 0.5 to 7 μm and the narrower size from 1 to 3 μm) enhanced the biocompatibility of GO. In both cases, the functionalization of GO with iron oxide NPs resulted in the higher biocompatibility of nanocomposite. The effect of GO-Fe_3_O_4_ on L929 cells was similar for B-GO-Fe_3_O_4_ (1–3 μm) and N-GO-Fe_3_O_4_ (0.5–7 μm), the only difference was noticed for the highest nanocomposite concentration (100 μgmL^−1^). L929 exposure to N-GO-Fe_3_O_4_ resulted in lower relative cell viability (~80%) in comparison with the cells incubated with B-GO-Fe_3_O_4_ (at concentration 100 μgmL^−1^) and the control culture [28]. Similarly, the exposure of human cell lines and zebrafish embryos to rGO/Fe_3_O_4_ in another experiment presented by Karthika et al. also confirmed the high biocompatibility of the tested nanocomposite [29]. The relative cell viability and activity was confirmed by MTT analyses [29]. A report by Chaudhari et al. described the effects of GO-IOI (graphene/magnetite nanocomposites fabricated in situ), GO-IOF (GO-IO prepared using ferrofluid) and GO-IOFA on the MCF-7 breast cancer cell line using the MTT method [30]. Chaudhari et al. found higher cytotoxicity for GO-IOFA than GO-IOIA and pure ANS and confirmed that complex nanoparticles affect cellular metabolism more effectively [30]. Magnetic composites were also evaluated by Shen et al. [31] Ultrafine graphene oxide-iron oxide (uGO@Fe_3_O_4_ NPs) was incubated with human hepatocyte cancer cells (HepG2) and normal hepatocytes (L02). The results showed that even at a high Fe concentration (400 μgmL^−1^) the uGO@Fe_3_O_4_ NPs show relatively high biocompatibility for both types of cell lines [31]. The cytotoxic effects of GO-Fe_2_O_3_ nanocomposites were analyzed by Gade et al. [32]. It was found that cells (caprine Wharton’s jelly derived mesenchymal stem cells, WJ-MSCs) exposed to 50 μgmL^−1^ and 100 μgmL^−1^ GO-Fe_2_O_3_ exhibited morphological alternation and decreased relative cell viability. The CFU (colony forming unit) assay confirmed the cytotoxic effect on WJ-MSC cells at 50 μgmL^−1^ and 100 μgmL^−1^ in comparison to the control and 10 μgmL^−1^ treatment group [32].

The results obtained in cell cultures treated with hydroxycamptothecin and its derivate in many studies stress the importance of using the mentioned drugs in anticancer therapy. HCPT displays strong anti-tumor effects with fewer side effects and can be widely used in a board spectrum of cancer treatments. Its activity is based on topoisomerase I (TOP I) inhibition. The TOP I is a ubiquitous enzyme that is involved in the regulation of DNA topology in such processes as replication, recombination, and transcription. Topoisomerase I allows for the formation of a phosphodiester bond between the enzyme and DNA. HCPT intercalates into the TOP I-DNA complex [33] resulting in a collision between the replication fork and the TOP I cleavable complex and the irreversible arrest of the replication fork (during S-phase) and replication-mediated DNA double-strand breaks (DSBs) [34,35]. HCPT also can create a collision between the elongating RNA polymerase complex and the TOP I cleavable complex that blocks transcription [35]. It was also found that HCPT displays redox properties [34,36]. The formed DSBs can lead to apoptotic cell death [34].

In our study, the effect of HCPT on MCF-7 was also evaluated. The MCF-7 cell cultures were treated with a hydroxycamptothecin solution in the concentration range of 0.0 to 50.0 µgmL^−1^ (Figure 7b). The results of theWST-1 assay showed a reduction in relative cell viability to 30% compared to free-growing cells at all concentrations tested. The data from the LDH assay demonstrated a low level of leakage (~10–15%) for all doses of HCPT used in the experiment. A high reduction in the proliferative activity and growth rate of MCF-7 cells were recorded for CPT6 (CPT analogue, camptothecin-20(s)-*O*-(2-pyrazolyl-1)acetic ester) by Chu at al. [36]. Another CPT analogue (CPT417, camptothecin-20-(S)-4-fluorophenoxy-acetic acid ester) also showed strong anti-tumor activity in the experiments involving both in vitro and in vivo models in the study by Yount et al. [37] CPT417 was shown to effectively inhibit cell growth and proliferation [38]. The same trend was also observed by Camacho et al. when a combination of CPT and DOX conjugated to a natural water-soluble biopolymer, hyaluronic acid (HA), were administered to the murine 4T1 breast cancer in vivo model [39]. A significant tumor reduction was recorded when a combination of these drugs was used [40]. Another form of HCPT—the hydroxycamptothecin—emulsion (HCPT-E) also showed antiproliferative activity. HeLa cells displayed changes in morphology and cells underwent apoptosis after 72 h of the exhibition to HCPT-E at the concentration at the range from 0.5 to 10 μM [41]. Bao et al. reported that gold nanoparticles (AuNPs) conjugated with HCPT were tested for cytotoxic effect in vitro and in vivo [41]. The cytotoxic effect was tested on the MDA-MB-231 cell line and the HCPT-AuNPs exhibited the highest cytotoxic effect with the mean diameter of ~50 nm at a concentration of 100 and 200 μgmL^−1^ [42].

### 2.6. The Effect of the GO-Fe_3_O_4_ Loaded with HCPT on MCF-7 Cells

Prior to analyzing the synergistic effect of GO-Fe_3_O_4_-HCPT and RMF on MCF-7 cells, the cytotoxicity of c-GO-Fe_3_O_4_-HCPT and nc-GO-Fe_3_O_4_-HCPT was evaluated. Both c-GO-Fe_3_O_4_-HCPT and nc-GO-Fe_3_O_4_-HCPT strongly affected cell viability and cellular activity in a dose-dependent manner. The strongest reduction in the relative cell viability (data obtained from WST-1) was recorded at a concentration of 50.0 µgmL^−1^ (Figure 8). Even the cells exposed to a lower dose of the nanocomposites exhibited reduced relative viability by ~85–90%. The difference between the cytotoxicity of c-GO-Fe_3_O_4_-HCPT and nc-GO-Fe_3_O_4_-HCPT was rather insignificant. The minimal effect on the viability of the MCF-7 cells was found in the samples exposed to nanocomposites for 48 h and 72 h. Slightly lower viability was recorded for the samples incubated for 72 h. The LDH and NR assays confirmed the data obtained from the WST-1 assay. The LDH leakage demonstrated higher cytotoxicity of both tested GO-Fe_3_O_4_ nanocomposites loaded with HCPT. However, it could be observed that c-GO-Fe_3_O_4_-HCPT induced higher LDH leakage than nc-GO-Fe_3_O_4_-HCPT after a 48 h incubation period. When the MCF-7 cell cultures were incubated with both nanocomposites for 72 h, the difference between the samples was very small. The NR assay also confirmed the tendency observed in the WST-1 assay. The neutral red uptake assay presented a slightly higher reduction in viability for both tested nanocomposites (c-GO-Fe_3_O_4_-HCPT and nc-GO-Fe_3_O_4_-HCPT) in comparison with WST-1 results. Shen et al. prepared superparamagnetic and dual drug-loaded nanocomposite (with two anti-cancer drugs—camptothecin (CPT) and methotrexate (MTX)) that were tested using an in vitro model—the HepG2 cell line—as well as an in vivo model of S-180 sarcoma-bearing Balb/c mice [38]. Shen et al. concluded that the combination of chemotherapy based on the nanocomposite-mediated dual drug with photothermal therapy had a remarkable synergic therapeutic potential against drug-resistant tumors [38]. Qu and co-workers synthesized and analyzed magnetic Fe_3_O_4_ nanoparticles coated with chitosan (CS) as a nano-sized carrier for HCPT. The CS-Fe_3_O_4_ NPs were conjugated with PEG to improve their biocompatibility and the obtained novel nanomaterial was tested by the exposure of the HepG2 cell line. The CS- Fe_3_O_4_ did not evoke a cytotoxic effect. The PEG-CS-Fe_3_O_4_ conjugated with HCPT exhibited antitumor activity with the highest inhibition of cell proliferation at a concentration of 15–20 μgmL^−1^ of HCPT loaded onto PEG-CS-Fe_3_O_4_ NPs. The presented study also confirmed the high saturated magnetization of PEG-CS-Fe_3_O_4_ NPs, thus it can be found as a novel and promising tool for a magnetic targeting drug delivery system in cancer therapy [43]. In other experiments by Zhang et al., oleic acid-Triton X-100-coated Fe_3_O_4_ nanoparticles loaded with HCPT were tested against human lung cancer cells (HCC827 cell line). Zhang et al. highlighted the potential of HCPT-NPs as a chemotherapeutic agent in anti-cancer drug therapy [44]. As concluded by Siafaka et al., nanoparticles, due to their lower size and high surface volume ratio, can be ideal platforms for drug delivery [42]. Native features such as toxicity and targetability can be improved by functionalization processes and can be used as an effective and relatively safe tool for anti-cancer treatments [42].

### 2.7. The Rotating Magnetic Field and Its Effect on the Cellular Activity of MCF-7

The effect of a rotating magnetic field (at different magnetic inductions (*B*) ranging from 1.23 to 10.06 mT) on the cell viability of the MCF-7 cultures was also analyzed in our study. The MCF-7 cell line was exposed to an RMF for 8 h, after which half of the samples were briefly analyzed. The second batch of cultured samples was analyzed 24 h after RMF exposure. The effect of the different parameters (*B* from 1.23 to 10.06 mT) of the applied RMF on the MCF-7 cell line was dose-dependent (Figure 9). The MCF-7 cell cultures exposed to the highest magnetic induction (10.06 mT, 50 Hz) exhibited a lower succinate-tetrazolium reductase system activity (~20%) than the cells exposed to 1.23 mT (~70%). The LDH assay results confirmed the data obtained in the WST-1 assay. LDH leakage was the lowest for a magnetic field of 1.23 mT (~20–25%). The highest LDH leakage was observed for cell cultures incubated in the magnetic field of 10.06 mT (~55%). Neutral red uptake verified the results obtained in WST-1 and LDH assays. The results for RMF treatment for 24 h demonstrated a similar tendency, but the relative cell viability (~40–75%) was higher. This phenomenon might be explained by the mechanism of temporary membrane permeabilization, as suggested by Liu et al. [9]. Also, LDH leakage was reduced. The MCF-7 cultures exposed to 3.95–10.06 mT magnetic fields for 24 h showed a lower rate of LDH leakage. The highest LDH leakage was observed for the cultures exposed to 1.23–2.36 mT magnetic fields. The NR results indicated lower neutral red uptake after 24 h of RMF exposure. The observed changes in cellular metabolism might be determined by cell membrane poration and ablation of cells incubated under RMF [21]. Liu et al. found that membrane permeabilization is repairable when cells are exposed to a weak RMF [9]. The magnetopores can reseal in the membrane, similar to what was documented for cell electroporation [9].

### 2.8. Potential Co-Effect of GO-Fe_3_O_4_ Nanoparticles Loaded with HCPT and RMF on the Viability of MCF-7 Cells

Finally, the synergistic effect of GO-Fe_3_O_4_ loaded with HCPT and rotating magnetic field was examined for the level of succinate-tetrazolium reductase system (EC 1.3.99.1) activity obtained in the WST-1 assay. It was found that MCF-7 cells exposed to the lowest concentration—3.125 µgmL^−1^ of c-GO-Fe_3_O_4_-HCPT were not significantly affected (Figure 10 and Appendix A). The highest relative viability was recorded for the concentration of 3.125 µgmL^−1^ of c-GO-Fe_3_O_4_-HCPT and a magnetic field of 1.23, 2.36, 3.95 and 10.06 mT. Interestingly, the highest mitochondrial activity under a magnetic induction of 1.57 mT was observed for 6.25 µgmL^−1^. In other concentrations of tested c-GO-Fe_3_O_4_-HCPT and RMF (in the range of 6.58–10.06 mT), cell viability was reduced by ~80–98%. Twenty-four hours after RMF exposure, the viability of cells was higher, while the co-effect of the nanomaterials and the magnetic field was temporal, due to the membrane cell poration obtained by RMF. Moreover, to characterize the effect of two analyzed factors and determine the potential interaction between the concentration of nanomaterials decorated with HCPT and magnetic induction values of RMF on adenocarcinoma MCF-7 cells, 3D response surface graphs and the Pareto charts were prepared (Figure 10, Figure 11, Figure 12, Figure 13, Figure 14 and Figure 15). In the Pareto charts for MCF-7 cell viability (WST-1 results, Figure 10) it can be noticed that the concentration of c-GO-Fe_3_O_4_-HCPT (the value which crosses the reference line at 0.05) is the factor with the highest influence (statistically significant). The same tendency was observed for results from the analysis performed directly after the RMF exposure (Figure 10b) as well for results obtained 24 h later (Figure 10d).

The results obtained for MCF-7 cells exposed to c-GO-Fe_3_O_4_-HCPT in LDH leakage and neutral red assays also confirmed the trend observed in the WST-1 assay. The highest LDH leakage was observed at a 50.0 µgmL^−1^ concentration and 10.06 mT magnetic induction (Figure 11 and Appendix A). This observation suggested a synergic effect of the nanocomposites and RMF. For other variants of nanocomposite concentrations and the applied magnetic field, LDH leakage was similar, and the reduction of cell viability was detected in the range of 55–65% for samples analyzed directly after exposure to RMF and those analyzed 24 h later. The NR uptake assay resulted in lower values in all examined combinations of c-GO-Fe_3_O_4_-HCPT and RMF (Figure 12 and Appendix A). The results obtained after a 24 h of RMF exposure showed a lower reduction of cell viability in the NR assay, especially for samples exposed to RMF in the range of 1.23–1.57 mT. The results obtained in WST-1, LDH and NR assays demonstrated the cytotoxic co-effect of **n**c-GO-Fe_3_O_4_-HCPT and RMF in cancer cell cultures.

The Pareto charts for the LDH leakage results (Figure 11) demonstrated a different tendency than in the WST-1 assay results. In that case, not only the concentration of c-GO-Fe_3_O_4_-HCPT is the factor with the highest statistically significant influence, but also magnetic induction (L—the linear value which crosses the reference line at 0.05) observed for the results from the analysis performed directly after RMF exposure (Figure 11b). For the results obtained 24 h later, only the concentration of c-GO-Fe_3_O_4_-HCPT was crucial (Figure 11d). For the neutral red analysis, the Pareto charts (Figure 12) indicate the same tendency as was observed for the WST-1 results. The cells’ ability to accumulate neutral red dye was affected mainly by the c-GO-Fe_3_O_4_-HCPT concentration (Figure 12b,d) in both cases of the preformed analysis (directly after RMF exposure and 24 h later).

In the case of nc-GO-Fe_3_O_4_-HCPT, the relative viability was highly reduced to 10–20% compared to free-growing cells (Figure 13 and Appendix A). The Pareto charts (Figure 13b,d) demonstrate that both factors—the nc-GO-Fe_3_O_4_-HCPT concentration and magnetic induction (of RMF) evoked the highest changes in MCF-7 cell viability (for results obtained directly after an 8 h incubation period in RMF).

The LDH and neutral red assay results demonstrated a strong reduction in the relative cell viability. The results recorded directly after the RMF treatment indicated that the c-GO-Fe_3_O_4_-HCPT nanomaterial affected cell viability more strongly than nc-GO-Fe_3_O_4_-HCPT, but the measurement performed 24 h after the experiment showed the higher cytotoxic effect of nc-GO-Fe_3_O_4_-HCPT and RMF on the MCF-7 cell line rather than c-GO-Fe_3_O_4_-HCPT (Figure 14, Figure 15 and Appendix A). Moreover, results from the LDH assay presented on Pareto charts indicate that in both cases—the **n**c-GO-Fe_3_O_4_-HCPT concentration and RMF (values which cross the reference line at 0.05) affected the cellular response and evoked the LDH leakage effect (Figure 14b,d).

The effect of nc-GO-Fe_3_O_4_-HCPT and an RMF on the cells was not only preserved after 24 h but became even more profound. In NRU analysis, the highest influence (statistically significant) was found for the nc-GO-Fe_3_O_4_-HCPT concentration (Figure 15b,d). These findings have shown that both tested nanomaterials: c-GO-Fe_3_O_4_-HCPT and nc-GO-Fe_3_O_4_-HCPT, as well as the RMF, effectively affected cellular metabolism and cell viability. However, it could be observed that nc-GO-Fe_3_O_4_-HCPT induced a greater reduction in mitochondrial activity than c-GO-Fe_3_O_4_-HCPT during and after RMF exposure. This suggests that it is possible to obtain effective anti-tumor activity when cancer cells are exposed to the multifactorial treatment. For example, Majeed et al. described the effect of the AC magnetic field (AMF) and nanomaterials (Fe_3_O_4_-MNPs and two types of Fe_3_O_4_-SiO_2_) on HeLa cells [45]. In the latter study, cell viability was significantly reduced as a result of combined treatment. Our results confirmed the findings presented by Majedd et al. [46], who found that in the absence of an AC (alternating current) magnetic field, the biocompatibility of nanoparticles was higher than in the samples exposed to a magnetic field; similar results were obtained by Gkanas [47]. Liu et al. also concluded that membrane cell poration obtained via RMF and combined with nanomaterial treatment had the most effective anti-cancer activity [9].

The effect of an RMF on biological systems involves different mechanisms that can be defined by physical and biological concepts. The magnetic field can penetrate not only the cellular membrane but also the cell body. According to Faraday’s law, biological tissues are diamagnetic, thus direct mechanical actions, e.g., transient rotational motions of substances proceed to achieve minimum energy states, can be excluded (their magnetic susceptibilities are close to a vacuum) [48]. On the other hand, the mechanical relative motions of a conducting medium could induce changes due to surface charges on the cell membranes (e.g., changes in membrane potential, re-orientation of the diamagnetic membrane phospholipids, sodium and calcium content, potassium- and calcium-efflux) [48,49]. An MF affects the reorganization of the electrostatically negatively charged actin filaments (changes in the cytoskeletal organization such as cellular shape, endoplasmic reticulum or mitotic apparatus) [49]; it also impacts the Brownian motions of ions and their transport through ion channels in cell membranes [48]. Hristov and Perez found that a magnetic field also affected the rate of ligand binding with the effect the signaling chain involving Ca^2+^ transport and growth factors [48]. The action of the magnetic field (MF) can be related to two processes—heat dissipation and mixing at the micro-level. The heating process—the ohmic heating of the volumes of the liquid resulted in heat dissipation [48]. The mixing process at the micro-level is related to the torques on the liquid volumes evoked by external MFs and eddy currents [48]. These two presented processes affect cellular metabolism and induce a heat-shock response related to the micro-level dynamo concept (MLD) [46,49,50]. The cellular response to heat-shock involves changes in transmembrane transport and receptor proteins on the cell surface [49]. It induces changes at the DNA level, including DNA damage or fragmentation, as well as affects protein denaturation in the nuclearmatrix. Cell cultures exposed to moderate heat can synthesize heat-shock proteins (HSP), which are involved in cell protection against heat damage (these genes play also a crucial role in normal cell processes in the absence of stressors). Intracellular HSP-synthesis increases as a response to heat and can determine thermotolerance, which is different for specific cell lines [46,48,50].

For example, Tokalov and Gutzeit reported that the expression of heat shock (HS) genes (such as *HSP27*, *HSP60*, *HSP70*, and *HSP90*) in human myeloid leukemia (HL-60) cell cultures was induced by extremely low-frequency electromagnetic fields (ELF-EMF) after a 30 min exposure [50].

## 3. Materials and Methods

### 3.1. Synthesis of Graphene Oxide (GO)

The modified Hummers method was used to obtain graphene oxide (GO) by graphite oxidation [51]. Briefly, graphite flakes (1.0 g) and of KmnO_4_ (6.0 g) were mixed in a round-bottom flask. Then, a mixture of two concentrated acids—H_2_SO_4_ (120 mL) and H_3_PO_4_ (15 mL) was slowly poured into the flask with powders. The reaction proceeds for 12 h (stirred using a magnetic stirrer and heated to 50 °C). Then, the reaction mixture was cooled down and 1 mL of H_2_O_2_ (30%) was slowly added. To remove metal ions and to reach pH 7.0, the mixture was purified with sequential washing and centrifugation with water, HCl aqueous solution (1:3) and ethanol. The obtained graphene oxide nanoplates were vacuum-dried for 12 h at 60 °C.

### 3.2. Synthesis of Magnetite Nanospheres (Fe_3_O_4_)

The magnetite nanoparticles (Fe_3_O_4_) were synthesized using the method described here [52]. Briefly, 20 mL of ethylene glycol (EG) and 400 mg of iron chloride (FeCl_3_) were mixed and sonicated to obtain the homogeneous dispersion. Then, sodium acetate (5.0 g) was added to the mixture and the dispersion was transferred into a sealed Teflon-lined stainless steel autoclave for 6 h at 200 °C. In the next step, the suspension was separated using a magnet and washed with ethanol and water. The resulting Fe_3_O_4_ nanospheres were vacuum dried at 100 °C. Similar method has been used e.g., by Xu et al. [53,54].

### 3.3. Synthesis of Graphene Oxide Decorated with Magnetite (Fe_3_O_4_) Nanoparticles

The functionalization of iron oxide with long-chain surfactants, polymers, dendrimers, inorganic silica and gold has been widely reported. However, the short-chain amino acid has certain advantages over the above-mentioned agents in terms of the usage in biomedicine due to their biocompatibility, low toxicity and lack of steric hindrances affecting binding affinity [55]. Therefore, glycine was chosen for functionalization as it is a short molecule with amino and carboxylate groups, easily adsorbed onto the iron oxide surface. Specifically, the carboxylate groups of glycine strongly coordinate to iron cations on the iron oxide surface [56], while the exterior amino groups can be then combined with carboxylate groups present in graphene oxide. The presence of –COOH groups in graphene oxide structure is a result of the interaction between graphite and strong oxidizing agents [57,58].

To obtain GO-Fe_3_O_4_ hybrid nanoparticles, Fe_3_O_4_ nanoparticles (20 mg) were dispersed in distilled water at the concentration of 0.5 mgmL^−1^ and ultrasonicated until a homogeneous dispersion. In the next step, the graphene oxide (20 mg) was exfoliated in H_2_O (60 mL) by ultrasonication to obtain a homogeneous graphene oxide suspension. Next, the carboxylic groups on graphene oxide nanoflakes surface were activated using N-hydroxysuccinimide (NHS; 8 mg) and 1-(3-dimethylaminopropyl)-3-ethylcarbodiimide (EDC; 10 mg). The prepared mixture of modified Fe_3_O_4_ and GO was stirred for 2 h. The obtained GO-Fe_3_O_4_ hybrid nanoparticles were centrifuged, washed several times with water and ethanol and then dried at a temperature of 100 °C.

### 3.4. Synthesis of Graphene Oxide-Fe_3_O_4_ Nanocomposites Covalently and Non-Covalently Conjugated with HCPT

Two synthesis approaches of graphene-magnetite based nanocomposites with hydroxycamptothecin (HCPT) were carried out. The synthesis of GO-Fe_3_O_4_ nanocomposite non-covalently conjugated with hydroxycamptothecin involved the physical adsorption of HCPT on the surface of nanocomposite through π-π interactions. Firstly, GO-Fe_3_O_4_ nanocomposite (7.0 mg) was ultrasonicated in distilled water (70 mL). And after that, GO-Fe_3_O_4_ was mixed with HCPT (5.0 mg) dispersed in dimethylsulfoxide (DMSO; 2 mL) and stirred for 24 h in the dark. Secondly, the newly synthesized product was thoroughly washed with dimethylformamide (DMF) to remove free substrates. The obtained sample was named nc-GO-Fe_3_O_4_-HCPT. The synthesis of the second nanocomposite was based on the chemical (covalent) bonding of modified HCPT with GO-Fe_3_O_4_. In the first step, GO-Fe_3_O_4_ (7.0 mg) was dispersed in distilled water (70 mL) and sonicated. Glycine solution was added to the dispersion and the mixture was stirred for 24 h. The semi-product was subsequently washed thoroughly with water. In the next step, hydroxycamptothecin was treated with 4-dimethylaminopyridine and succinic anhydride, according to the method reported by Wu et al. [15] to obtain succinate-based HCPT ester derivative with a carboxyl group in the structure [16,59]. To activate carboxyl groups the obtained d-HCPT was dispersed in DMF with NHS and EDC. GO-Fe_3_O_4_ was re-dispersed in DMF (used to purify the novel nanocomposite), added to d-HCPT and stirred in the dark at room temperature overnight. The synthesized sample was named c-GO-Fe_3_O_4_-HCPT.

### 3.5. Characterization of the Obtained Nanomaterials

The morphology of the samples was explored a high-resolution transmission electron microscopy (HRTEM) operated at 300 kV accelerating voltage (FEI Tecnai F30, Frequency Electronics Inc., Thermo Fisher Scientific, Waltham, MA, USA). The samples were dispersed in ethanol, sonicated for 10 min and small drops of the prepared dispersions were deposited on copper grids with a thin layer of carbon. The grids were dried at 60 °C. As-prepared samples were placed onto a holder.

Atomic force microscopy (AFM) using a contact-mode AFM (CM-AFM) (Nanoscope V Multimode 8, Bruker AXS, Mannheim, Germany) has been used to investigate the morphology, thickness and size of graphene oxide flakes. The information about the morphology of tested nanomaterials has been obtained using NanoScope Analysis Software v. 1.7 (Nanoscope V Multimode 8, Bruker AXS, Mannheim, Germany).

X-ray diffraction technique (X-ray diffractometer Philips X’Pert PRO, PANalytical Almelo, Holland) using Kα1 = 1.54056 Å was employed to examine the structure of the samples. The powders were pressed onto holders and the measurements were carried out in the range of 7–50° and 25–70°of 2θ for GO and the samples with Fe_3_O_4_ nanospheres, respectively. Collected spectra were analyzed with Philips X’Pert HighScore Plus software version 2.0.

IR absorption spectra of novel samples were analyzed using a Nicolet 6700 FTIR spectrometer (Thermo Nicolet Corp., Madison, WI, USA). Each peak present on the IR spectrum is associated with specific binding. Additional bands on the spectrum of c-GO-Fe_3_O_4_ can indicate the covalent bonds between nanocomposite and anticancer drug, hydroxycamptothecin (HCPT). The data were collected in the absorption mode in the range of 500–3750 cm^−1^ and analyzed by OMNIC Professional 7 software package (Thermo Fisher Scientific, Waltham, MA, USA).

### 3.6. Dispersion Stability of GO-Fe_3_O_4_-HCPT Covalently and Non-Covalently Conjugated with HCPT

The stability of the dispersions of the prepared nanocomposites in PBS solution was examined. To obtain nanocomposite suspensions at 1 mgml^−1^ concentrations, c-GO-Fe_3_O_4_-HCPT and nc-GO-Fe_3_O_4_-HCPT were added to phosphate-buffered solutions (1 mgmL^−1^). Next, the diluted suspensions of nanocomposites were sonicated to reach the following concentrations: 100 µgmL^−1^, 50 µgmL^−1^, 25 µgmL^−1^, 12.5 µgmL^−1^, 6.25 µgmL^−1^, and 3.125 µgmL^−1^. Dispersion stability was evaluated by monitoring UV/vis absorbance using the 325 nm wavelength changes at selected time points (1–120 h).

### 3.7. Determination of HCPT Loading and Release

The release behaviour of hydroxycamptothecin from nanocomposites was analyzed using the batch technique (at 37 °C). Approximately 2.0 mg of nc-GO-Fe_3_O_4_-HCPT and c-GO-Fe_3_O_4_-HCPT were introduced into the flasks pre-filled with 60 mL of PBS placed in a constant temperature bath. A magnetic stirrer was applied to receive homogenous dispersions. At predetermined times, each sample (0.5 mL) was taken from the solutions; the solutions and solid phase were separated using the centrifuge. The concentrations of HCPT after desorption in the supernatant solutions were examined using a UV–vis spectrophotometer (Thermo Scientific GENESYS 10S, Waltham, MA, USA) at λ max value of 364 nm. Analysis of UV spectra was performed using the VisionLite^TM^ 5 software (Thermo Scientific, Waltham, MA, USA) and OriginPro software v. 8.5 (OriginLab, Northampton, MA, USA). Each experiment was performed in triplicate and the results are given as average values with error bars.

The amount of HCPT loaded into the nanocomposites was determined by UV–vis spectroscopy using the method described by Tyner et al. [60]. 10 mg of each of the nanocomposites was dissolved in 10 mL HCl solution (1M), followed by addition of 40 mL of ethanol. The absorbance of the solutions was determined at λ = 364 nm and the concentrations of HCPT was calculated by regression analysis according to the standard curve obtained from a series of the standard solution of HCPT. The amount of loaded HCPT was calculated according to the ratio of HCPT mass in the solution and the mass of the used nanocomposite sample.

### 3.8. Cell Culture and Treatment Conditions

The MCF-7 (human breast adenocarcinoma) cell line was purchased from American Type Culture Collection (ATCC) C (MCF-7 cell line, HTB-22™, ATCC^®^, Manassas, VA, USA obtained in 2018). In all experiments were performed below 13 passages and cell line authentication was performed by ATCC using STR profiling. MCF-7 human breast adenocarcinoma was seeded in 96-well culture microplates (Corning Inc., New York, NY, USA) at an initial density of 4 × 10^3^ cells per well and it was cultured under defined conditions at 37 °C in 5% CO_2_ and 95% humidity. DMEM medium (Dulbecco’s Modified Eagle Medium, High Glucose, PAA, Pasching, Austria) was supplemented with 10% heat-inactivated fetal bovine serum (FBS, PAA, Pasching, Austria), 2 mM L-glutamine (Sigma-Aldrich, St. Louis, MO, USA), streptomycin-penicillin (50 IUmL^−1^; Sigma-Aldrich, St. Louis, MO, USA) and 10 mM HEPES (Sigma-Aldrich, St. Louis, MO, USA) and was used to maintain cell cultures. The two tested Fe_3_O_4_-graphene oxide nanocomposites loaded with hydroxycamptothecin (c-GO–Fe_3_O_4_-HCPT, and nc-GO-Fe_3_O_4_-HCPT, respectively) were added to cell cultures 24 h after from seeding. The final concentrations of the NPs in the culture medium were: 3.125, 6.25, 12.5, 25.0 and 50.0 µgmL^−1^, respectively. The MCF-7 cells were exposed to nanomaterials for 48 h. Additionally, the effect of GO–Fe_3_O_4_ and HCPT not conjugated with NPs were analyzed.

### 3.9. The Rotating Magnetic Field Effect on Cells Relative Viability

After 48 h incubation with nanocomposites, MCF-7 cell cultures were placed in the generator of the rotating magnetic field (RMF). The RMF generator used in the presented study was constructed of the cylindrical glass container (water bath incubator) in the centre of the coil. The RMF was generated by the 3-phase stator of the squirrel cage induction motor. The three-phase windings of this stator were displaced from each other by 120°. These windings were supplied by a balanced three-phase ac supply. The three-phase currents flow simultaneously through the windings. An RMF, which emerges from the superposition of three, 120° out of phase magnetic fields, has a constant intensity over time while it changes its direction continuously at any point of the domain. The more detailed description of the generated magnetic field and the applied apparatus was given by Rakoczy (2013) [10]. The MCF-7 cells were exposed to the RMF with different magnetic induction values in the range from 1.23 to 10.06 mT and a frequency of 50 Hz for 8 h. All experiments were conducted in a temperature-controlled condition (37 °C ± 0.5 °C). Additionally, the effect of RMF in the absence of GO–Fe_3_O_4_-HCPT nanocomposites was also determined.

### 3.10. Observation of MCF-7 Cell Line Culture Morphology

After an 8 h combined exposure to RMF (at different magnetic inductions—1.23, 1.57, 2.36, 3.95, 6.58 and 10.06 mT) and c-GO-Fe_3_O_4_-HCPT/nc-GO-Fe_3_O_4_-HCPT at different concentrations, the morphology of MCF-7 cells was observed using an inverted phase-contrast microscope (Nikon TS-100 microscope, NIS Elements F Package, camera Nikon DS-Fi1, Nikon, Melville, NY, USA) at 400 × magnification.

### 3.11. WST-1 Cytotoxicity Assay

The potential co-effect of nanomaterials and rotating magnetic field on the MCF-7 cell line were examined using Cell Proliferation Reagent WST-1 (Roche Applied Science, Mannheim, Germany) to determine the relative mitochondrial cell activity after incubation with nanomaterials and an 8 h exposure to RMF.

In this experiment, 20 µL of WST-1 solution was added to each well of 96-well microplate and incubated for 60 min at 37 °C. Then, the absorbance values at 450 nm (with a reference wavelength at 630 nm) were measured with a Sunrise Absorbance Reader (Sunrise, Männedorf, Switzerland) equipped with the Magellan Standard Software version 7.2 (Tecan, Männedorf, Switzerland). The cell cultures maintained in complete DMEM medium without nanocomposites served as a negative control. The interaction between the tested nanomaterials and WST-1 reagents were examined using the suspension of nanomaterials at different concentrations in complete DMEM medium incubated for 48 h in standard culture conditions in the absence of cells [21].

### 3.12. Lactate Dehydrogenase Leaking Assay

The cytotoxicity of nanocomposites and rotating magnetic field were determined using LDH CytoTox 96^®^ Non-Radioactive Cytotoxicity Assay (Promega, Madison, WI, USA). Firstly, after the experimental treatment, the plates were centrifuged at 240× *g* for 4 min and the supernatant was transferred into new 96-well plates. Then, the transferred supernatant was mixed with Substrate Mix (Promega, Madison, WI, USA) and the plates were incubated for 30 min at room temperature (light-protected). Finally, Stop Solution (Promega, Madison, WI, USA) was added to the plates and the absorbance at 490 nm was measured using a spectrophotometer reader (Sunrise, Tecan, Männedorf, Switzerland) equipped with the Magellan Standard Software version 7.2 (Sunrise, Tecan, Männedorf, Switzerland). The nanocomposites—LDH assay components interaction was determined using different concentrations of the nanomaterials incomplete DMEM medium in the absence of cells [21,61].

### 3.13. Neutral Red Uptake Assay

Briefly, the medium of MCF-7 cells incubated for 48 h and additional 8 h exposure to RMF was collected and the cells were washed twice with phosphate-buffered saline (PBS) (Gibco, Langley, OK, USA). Fresh culture medium with 10% neutral red dye (In Vitro Toxicology Assay Kit, Neutral Red based, Sigma-Aldrich, St. Louis, MO, USA) solution was added to the cells and were incubated at 37 °C, 5% CO_2_ for another 3 h. Then, the cell cultures were washed twice with PBS and Neutral Red Assay Solubilization Solution (Sigma-Aldrich, St. Louis, MO, USA) was added to the microplates. The cell cultures were incubated for 10 min at room temperature, then were gently stirred for 10 min. The absorbance measurement was performed at a wavelength of 540 nm (background absorbance of Multiwall plates at 690 nm) using a microplate reader (Tecan, Männedorf, Switzerland) equipped with the Magellan Standard Software version 7.2 (Tecan, Männedorf, Switzerland) [21].

### 3.14. Statistical Analysis

The data analyzed in this study (WST-1, LDH, and NR assays) were showed as mean values ± standard deviation (SD) and analyzed using ANOVA. Duncan’s new multiple range test (MRT) was used for multiple comparisons of means (post hoc analysis) obtained from cell cultures exposed to graphene oxide, HCPT and RMF. The *p*-values ˂ 0.05 were considered significant (letters from a to g, Appendix A). The means obtained for c-GO-Fe_3_O_4_-HCPT and nc-GO-Fe_3_O_4_-HCPT were analyzed in single-step multiple comparisons using post hoc Tukey’s range test. Differences were considered significant at *p* < 0.05 (letters from a to g, Appendix A). One-way repeated measures analysis of variance (rmANOVA) was used to determine the effects of magnetic induction, nanomaterials concentration and time (data obtained after RMF exposure and 24 h later) on MCF-7 cell culture.

Statistical analysis was performed using the STATISTICA 12.5 software (StatSoft Inc., Tulsa, OK, USA).

## 4. Conclusions

In summary, the cytotoxicity of c-GO-Fe_3_O_4_-HCPT and nc-GO-Fe_3_O_4_-HCPT and a rotating magnetic field was evaluated in this study using cellular metabolic assays. The results show that the anti-tumor activity can be enhanced by exposing the cells incubated with c-GO-Fe_3_O_4_-HCPT or nc-GO-Fe_3_O_4_-HCPT to a rotating magnetic field. Both nanocomposites exhibited a cytotoxic effect in a human breast adenocarcinoma cell line (MCF-7) in the presence of an RMF, however, MCF-7 cells were more significantly affected by nc-GO-Fe_3_O_4_-HCPT. It was found that the concentration of nc-GO-Fe_3_O_4_-HCPT and the RMF have the highest statistically significant influence on MCF-7 cells viability. As presented above, HCPT, GO-Fe_3_O_4_, and an RMF did not affect human breast adenocarcinoma cells as much as the complex treatment. Therefore, the presented results have potential clinical applications, but still, more in-depth analyses need to be performed. Further investigations such as drug release experiments in the presence of an RMF will be performed in the future.

## Figures and Tables

**Figure 1 ijms-25-00930-f001:**
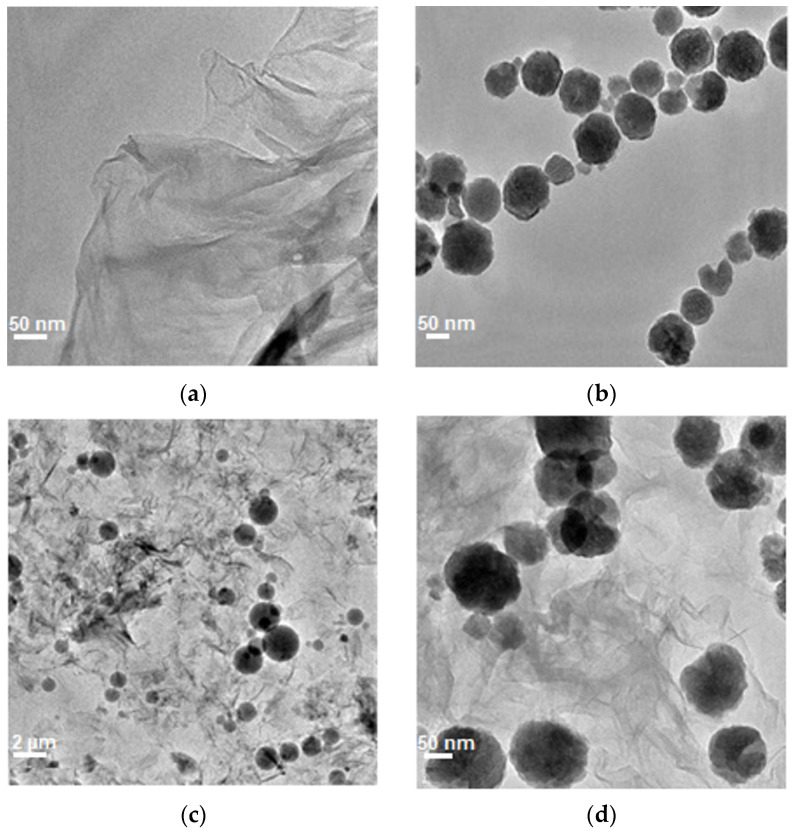
TEM images of GO (**a**), Fe_3_O_4_ nanospheres (**b**) and GO-Fe_3_O_4_ nanocomposite at a different magnification of the (**c**,**d**).

**Figure 2 ijms-25-00930-f002:**
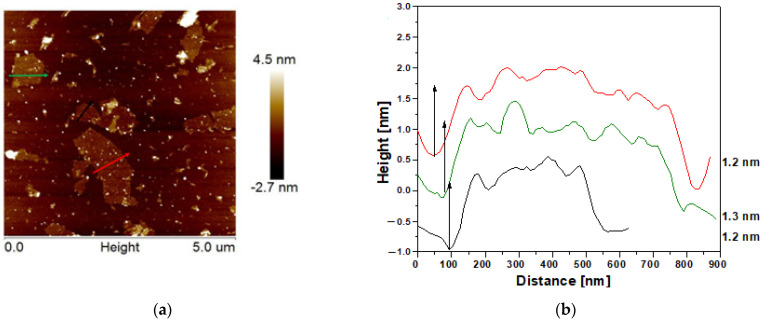
AFM image of GO (**a**), height profiles of chosen flakes (**b**), the size distribution of GO flakes (**c**), the size distribution of Fe_3_O_4_ nanospheres (**d**).

**Figure 3 ijms-25-00930-f003:**
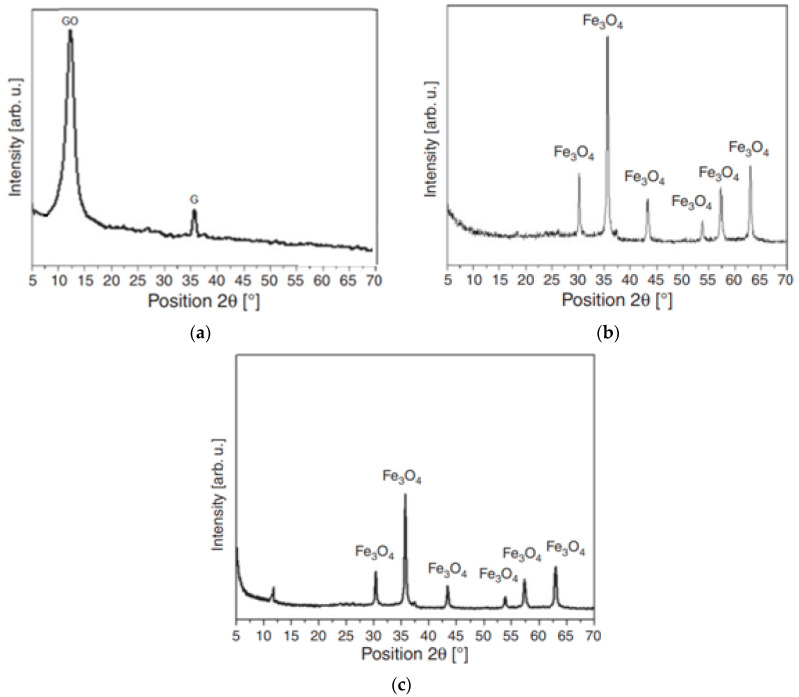
XRD spectra of GO (GO—graphene oxide, G—graphite) (**a**) and Fe_3_O_4_ nanospheres (**b**) and GO-Fe_3_O_4_ nanocomposite (**c**).

**Figure 4 ijms-25-00930-f004:**
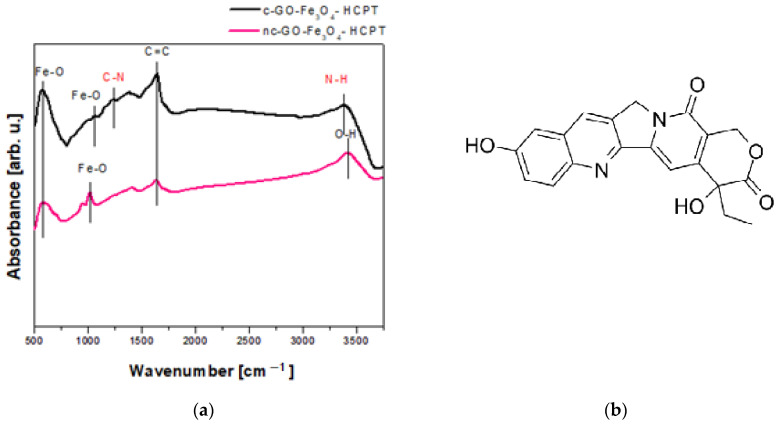
FTIR spectra of GO-Fe_3_O_4_ nanocomposite covalently and non-covalently loaded with HCPT (**a**) and the chemical structure of hydroxycamptothecin (**b**).

**Figure 5 ijms-25-00930-f005:**
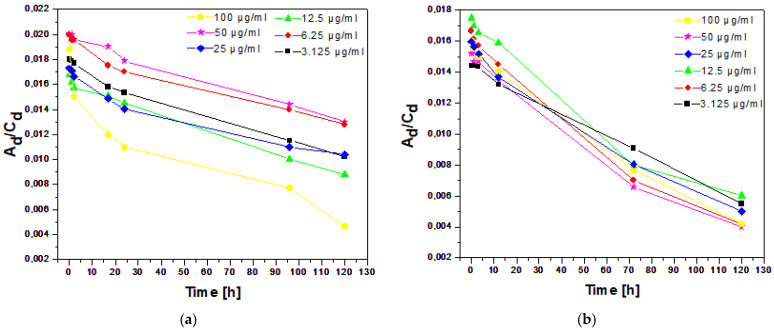
The change in the ratio of UV–vis absorbance (325 nm) of dispersion to dispersion concentration (A_d_/C_d_) in time as a measure of the stability of the dispersions of nc-GO-Fe_3_O_4_-HCPT (**a**) and c-GO-Fe_3_O_4_-HCPT (**b**) nanocomposites in PBS.

**Figure 6 ijms-25-00930-f006:**
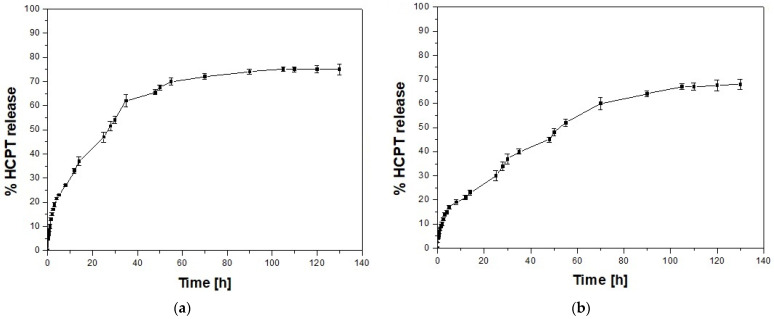
Comparative cumulative percent HCPT release from nc-GO-Fe_3_O_4_-HCPT (**a**) and c-GO-Fe_3_O_4_-HCPT (**b**) in PBS (pH 7.4, at 37 °C). The error bars are based on the standard deviation (SD) of three samples.

**Figure 7 ijms-25-00930-f007:**
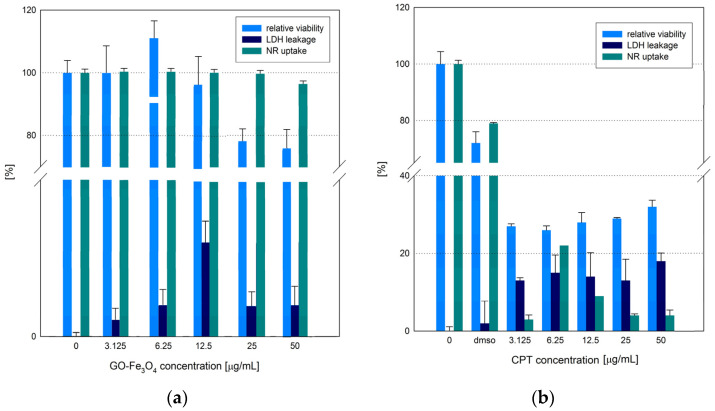
The cellular response (%) of MCF-7 to the GO-Fe_3_O_4_ nanocomposite (**a**) and HCPT (without exposure to RMF) (**b**) (all samples were compared to control samples, *p*-values ˂ 0.05 were considered significant and presented in Appendix A).

**Figure 8 ijms-25-00930-f008:**
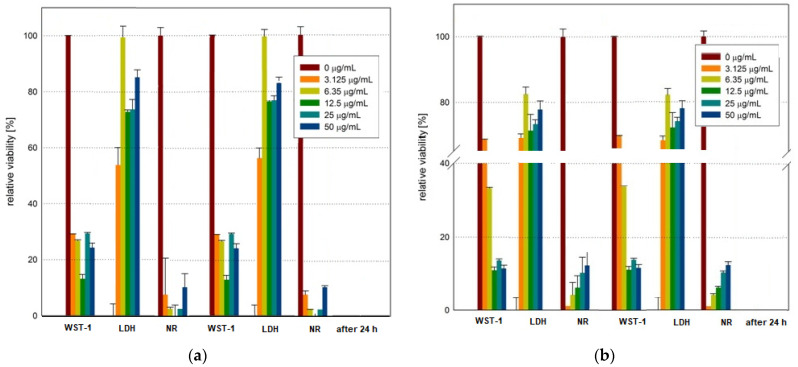
The cell viability (%) of MCF-7 after 48 h and 72 h incubation of c-GO-Fe_3_O_4_-HCPT (**a**) and nc-GO-Fe_3_O_4_-HCPT (**b**) (all samples were compared to control samples, *p*-values < 0.05 were considered significant and presented in Appendix A).

**Figure 9 ijms-25-00930-f009:**
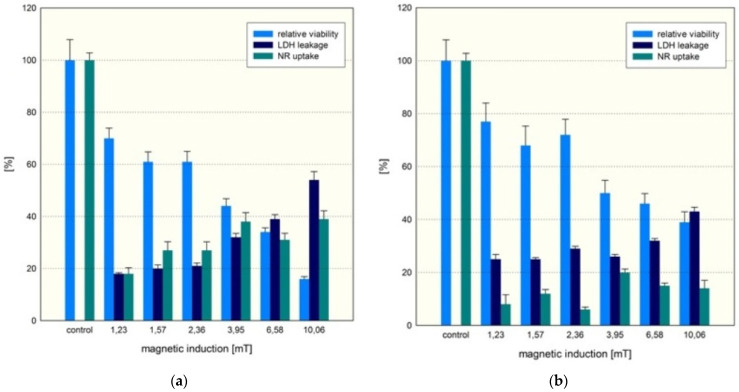
The relative viability of MCF-7 cells (**a**) after exposure to RMF (WST-1, LHD release and NR uptake assays) and (**b**) 24 h after exposure to RMF (all samples compared to “0” sample, *p*-values ˂ 0.05 are considered significant and are and presented in Appendix A).

**Figure 10 ijms-25-00930-f010:**
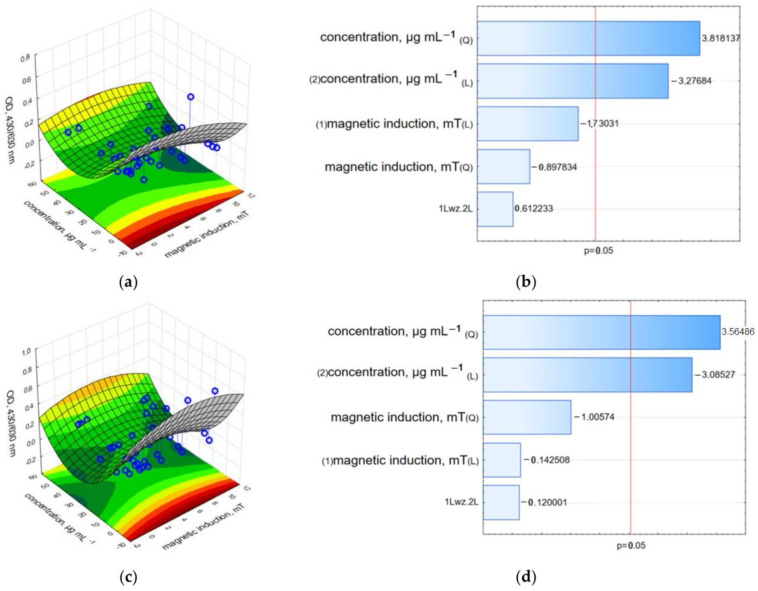
3D surface response graphs of the interaction between the concentration of c-GO-Fe_3_O_4_-HCPT and the magnetic induction of an RMF (**a**) and 24 h after exposure to an RMF (**c**) and Pareto charts (**b**,**d**) indicate the variables that were significant in MCF-7 cell viability (WST-1 assay) (*p* > 0.05; L—linear; Q—Quadratic).

**Figure 11 ijms-25-00930-f011:**
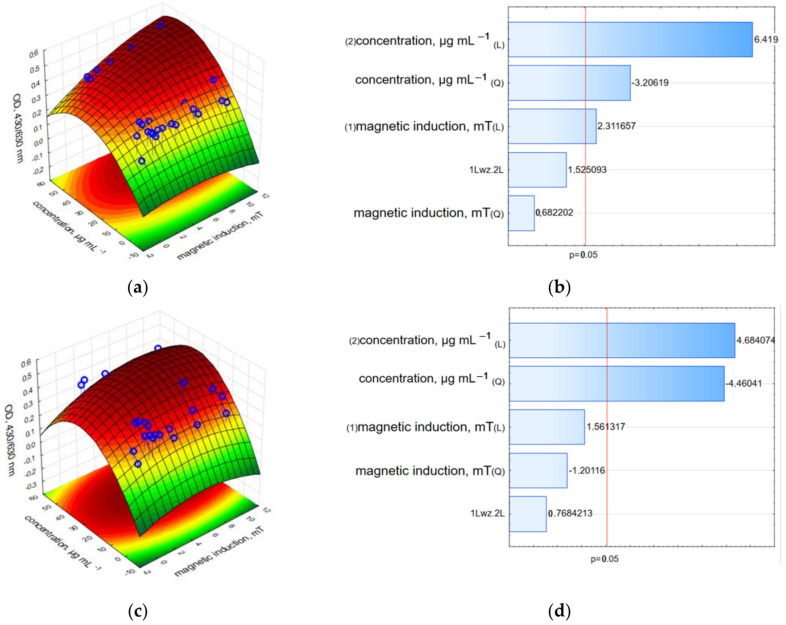
3D surface response graphs of the interaction between the concentration of c-GO-Fe_3_O_4_-HCPT and the magnetic induction of an RMF (**a**) and 24 h after exposure to an RMF (**c**) and Pareto charts (**b**,**d**) indicate the variables that were significant in MCF-7 cells viability (LDH leakage assay) (*p* > 0.05; L—linear; Q—Quadratic).

**Figure 12 ijms-25-00930-f012:**
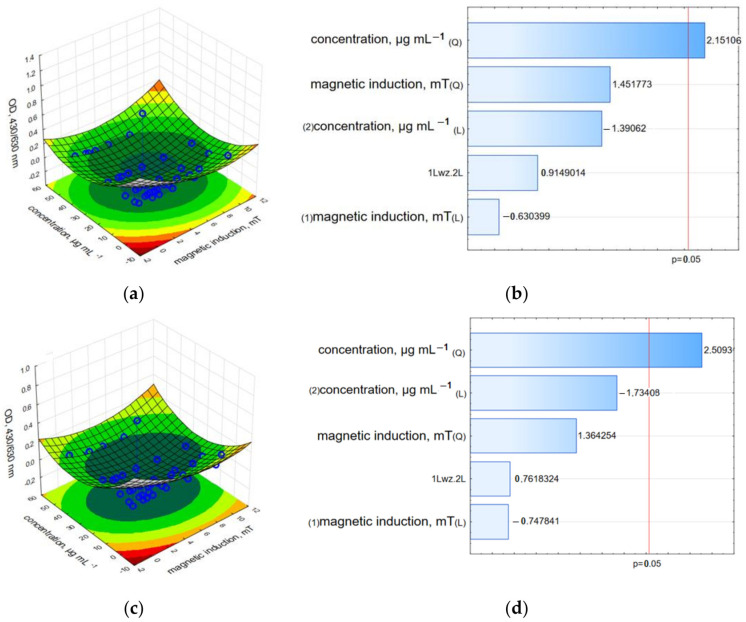
3D surface response graphs of the interaction between the concentration of c-GO-Fe_3_O_4_-HCPT and the magnetic induction of an RMF (**a**) and 24 h after exposure to an RMF (**c**) and Pareto charts (**b**,**d**) indicate the variables that were significant in MCF-7 cells viability (Neutral red uptake assay) (*p* > 0.05; L—linear; Q—Quadratic).

**Figure 13 ijms-25-00930-f013:**
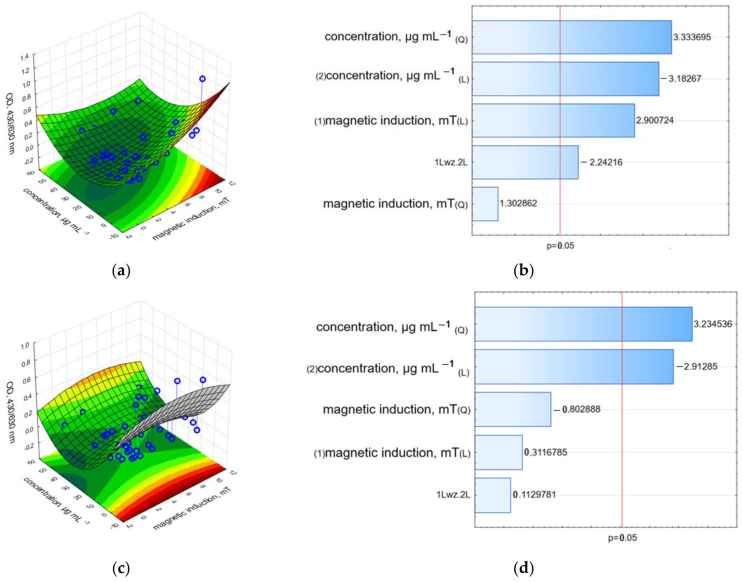
3D surface response graphs of the interaction between the concentration of nc-GO-Fe_3_O_4_-HCPT and the magnetic induction of an RMF (**a**) and 24 h after exposure to an RMF (**c**) and Pareto charts (**b**,**d**) the variables that were significant in MCF-7 cells viability (WST-1 assay) (*p* > 0.05; L—linear; Q—Quadratic).

**Figure 14 ijms-25-00930-f014:**
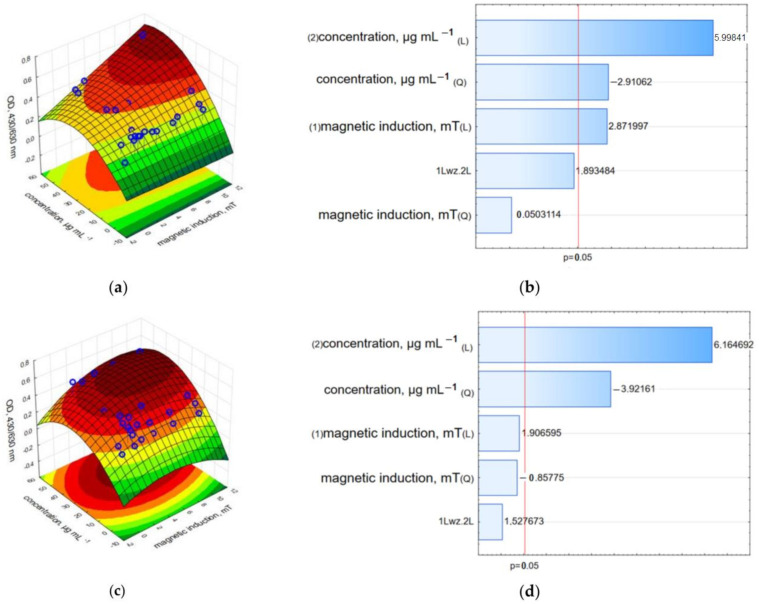
3D surface response graphs of the interaction between the concentration of nc-GO-Fe_3_O_4_-HCPT and the magnetic induction of an RMF (**a**) and 24 h after exposure to an RMF (**c**) and Pareto charts (**b**,**d**) indicate the variables that were significant in MCF-7 cells viability (LDH leakage assay) (*p* > 0.05; L—linear; Q—Quadratic).

**Figure 15 ijms-25-00930-f015:**
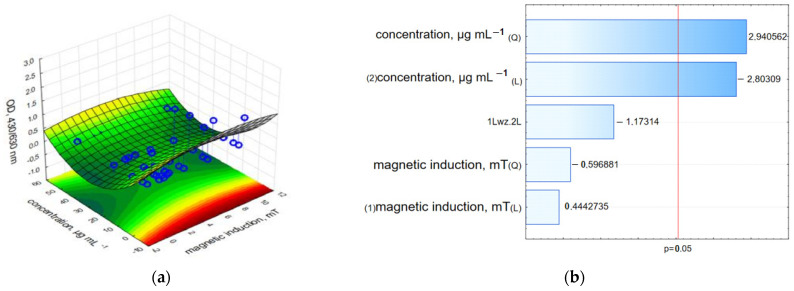
3D surface response graphs of the interaction between the concentration of c-GO-Fe_3_O_4_-HCPT and the magnetic induction of an RMF (**a**) and 24 h after exposure to an RMF (**c**) and Pareto charts (**b**,**d**) indicate the variables that were significant in MCF-7 cells viability (Neutral red uptake assay) (*p* > 0.05; L—linear; Q—Quadratic).

## Data Availability

Data available on request.

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
