# Peer review of "The Influence of Graphene Oxide-Fe3O4 Differently Conjugated with 10-Hydroxycampthotecin and a Rotating Magnetic Field on Adenocarcinoma Cells"

_ijms, 2024, doi:10.3390/ijms25020930_

Round 1

Reviewer 1 Report

Comments and Suggestions for Authors

Dear Authors

The MS entitled “The influence of graphene oxide – Fe3O4 differently conjugated with 10-hydroxycapthotecin and rotating magnetic field on adenocarcinoma cells” was thoroughly reviewed. it describes the synthesis of graphene oxide-Fe3O4 hybrids surfaces loaded with hydroxycamptothecin and its effect on breast cancer cell lines. Accordingly, the non-covalent bonded conjugates were active. The research is fine and updated.  A huge data has been sumarised in the MS. My suggestions are provided here.  

 1.      The abstract missing some core data that should suggest the effectiveness, in comparison to standard to validate the claim of authors. No data has been provided. The Graphene oxide -Fe2O3 characterization, rout of synthesis missing. Also, the loading and release ratio of drug missing from abstract. In my opinion, there is no need for the first sentence, from “An alternative…. Desirable” should be omitted.

2.      Key words should be rearranged and in order. graphene oxide - Fe3O4 nanocomposite rotating magnetic field (RMF) MCF-7 cell culture, cell viability biomedical application

3.      Line 86-100, also refer to recent GO dopped podophyllotoxin from literature https://doi.org/10.1021/acsomega.3c00888

4.      You can also do some amendments in your abstract or any other relevant literature.

5.      119-120. You termed these “novel” however, several reports have already been published. Kindly do justify the novelty here. How these hybrids are different from already used GO-magnetite surfaces?

6.      Provide chemical structure of hydroxycamptothecin at suitable place, near FTIR. As to affirm the presence of functionalities in this compound in hybrid material.

7.      In the FTIR, fig 5, why N-H bond is present? There is no NH in hydroxycmptothecin. Beside this, which atoms are making the covalent bond?  Is this with the Carbon of GO or the metal oxide functionality?

8.      If any co valent bonding arises, the nature of molecules changes. Are you supposing a new hybrid material where the drug is no longer in its original formulation?

9.      Section 2.3. HCPT loading and release. Needs extensive corrections. Not OK. Why not a simple release data, preferably, UV/Vis obtained?

10.   Line 219-223. Need serious discussion and avoid stating the future goals line 222-223 (should be part of conclusion).

11.   269-305. A lot of literature. Kindly make the discussion to the point and avoid provision of non-specific details.

12.   Line 350. Cytotoxicity was evaluated on which normal cell lines? Mention that. Also, if the hybrids are toxic to normal cell lines in even in very small doses, that could render the material unsafe.

13.   Line 367-394. Same comment as number 11. A lot of literature.

14.   Section: 3.4. line 633. Modifications in drug structure might resulted. Did you perform any NMR studies of resulted drug-derivatives?

15.   644. Correct °C. check all the MS for such type of mistakes.

16.   Line 652. How are you sure that there only Fe3O4 and no Fe2O3 in the nano form?

17.   655-665. No need of discussing the KBr disc preparation.

18.   Line 674. Why 325 nm and 365 nm (line 684) were selected for the measurements?

19.   All most all the references are very old. 

20. The work is very attractive and should be presented as attractive. 

Comments on the Quality of English Language

The English language needs revision. 

Author Response

Journal:                      International Journal of Molecular Science

Manuscript ID:           ijms-2776466R1

Manuscript title:        The influence of graphene oxide – Fe3O4 differently conjugated with 10-hydroxycapthotecin and rotating magnetic field on adenocarcinoma cells

Auhors:                      Magdalena Jedrzejczak-Silicka, Karolina SzymaÅ„ska, Ewa Mijowska, RafaÅ‚ Rakoczy

Reply for the Editors’ and Reviewers' comments

We would like to thank you very much for the valuable comments concerning our manuscript ” The influence of graphene oxide – Fe3O4 differently conjugated with 10-hydroxycapthotecin and rotating magnetic field on adenocarcinoma cells” (ijms-2776466) to International Journal of Molecular Sciences. We would like also to thank the Reviewers very much for their effort in the revision of the manuscript and all insightful comments and helpful suggestions which lead to a significant improvement of our manuscript. All suggestions were strong taken into consideration what surely has improved the value of the paper which reports, we believe, a very important message.

We hope that our corrections, answers and comments will be accepted by the Reviewers and the improved manuscript will be approved for publication in the International Journal of Molecular Science.

Editors and Reviewers #1 comments are in bold sans serif text.

Responses to Reviewers #1 comments are in un-bolded, serif text.

All the corrections in the manuscript have been marked in red color.

Reviewer #1:

The MS entitled “The influence of graphene oxide – Fe3O4 differently conjugated with 10-hydroxycapthotecin and rotating magnetic field on adenocarcinoma cells” was thoroughly reviewed. it describes the synthesis of graphene oxide-Fe3O4 hybrids surfaces loaded with hydroxycamptothecin and its effect on breast cancer cell lines. Accordingly, the non-covalent bonded conjugates were active. The research is fine and updated.  A huge data has been sumarised in the MS. My suggestions are provided here.  

  1. The abstract missing some core data that should suggest the effectiveness, in comparison to standard to validate the claim of authors. No data has been provided. The Graphene oxide -Fe2O3characterization, rout of synthesis missing. Also, the loading and release ratio of drug missing from abstract. In my opinion, there is no need for the first sentence, from “An alternative…. Desirable” should be omitted.

Answer:

Thank you for this remark. The abstract has been modified as was recommended (all corrections have been marked in red).

  1. Key words should be rearranged and in order. graphene oxide - Fe3O4 nanocomposite rotating magnetic field (RMF) MCF-7 cell culture, cell viability biomedical application

Answer:

Thank you for this remark. The key words have been modified as was recommended (all corrections have been marked in red).

  1. Line 86-100, also refer to recent GO dopped podophyllotoxin from literature https://doi.org/10.1021/acsomega.3c00888

Answer:

We appreciate your recommendation. According to Reviewer #1 suggestion the changes have been made (all corrections have been marked in red).

  1. You can also do some amendments in your abstract or any other relevant literature.

Answer:

Thank you for this remark. The manuscript has been modified as was recommended (all corrections have been marked in red).

  1. 119-120. You termed these “novel” however, several reports have already been published. Kindly do justify the novelty here. How these hybrids are different from already used GO-magnetite surfaces?

Answer:

Thank you for this remark. The result section has been modified as was recommended (all corrections have been marked in red). The novelty of the obtained hybrid is related with glycine that was chosen for functionalization as it is a short molecule with amino and carboxylate groups, easily adsorbed onto the iron oxide surface.

  1. Provide chemical structure of hydroxycamptothecin at suitable place, near FTIR. As to affirm the presence of functionalities in this compound in hybrid material.

Answer:

Thank you for this issue. The FTIR figure has been modified as was recommended (all corrections have been marked in red).

  1. In the FTIR, fig 5, why N-H bond is present? There is no NH in hydroxycmptothecin. Beside this, which atoms are making the covalent bond?  Is this with the Carbon of GO or the metal oxide functionality?

Answer:

The peaks derived from the Fe-O bond are present in both spectra. Additional peaks were observed in the c-GO-Fe3O4-HCPT spectrum (two additional peaks originated from the covalent bonding between GO and magnetite) in comparison to the nc-GO-Fe3O4-HCPT nanocomposite spectrum, where lack of chemical bonds between the nanoparticle carrier and the drug (HCPT) was observed. The c-GO-Fe3O4-HCPT spectrum is characterized by peaks at 1359 cm-1 derived from C-N bond and 3245 cm-1 arising from N-H (Figure 4). Therefore, the presence of the above-mentioned peaks proved a propitious covalent functionalization of the GO-Fe3O4 carrier with the anti-cancer drug. The remaining minor peaks in the range of 613 cm-1 to 1405 cm-1 in both spectra are associated with the presence of hydroxycamptothecin.

 The synthesis of the nanocomposite was based on the chemical (covalent) bonding of modified HCPT with GO-Fe3O4. In the first step, GO-Fe3O4 (7.0 mg) was dispersed in distilled water (70 ml) and sonicated. Glycine solution was added to the dispersion and the mixture was stirred for 24 h. The semi-product was subsequently washed thoroughly with water. In the next step, hydroxycamptothecin was treated with 4-dimethylaminopyridine and succinic anhydride, according to the method reported by Wu et al. to obtain succinate-based HCPT ester derivative with a carboxyl group in the structure. To activate carboxyl groups the obtained d-HCPT was dispersed in DMF with NHS and EDC. GO-Fe3O4 was re-dispersed in DMF (used to purify the novel nanocomposite), added to d-HCPT and stirred in the dark at room temperature overnight.

  1. If any covalent bonding arises, the nature of molecules changes. Are you supposing a new hybrid material where the drug is no longer in its original formulation?

Answer:

We appreciate the reviewer's insightful comment. It should be noticed that the presented paper has preliminary character. Consequently, use of that kind of research gives possibility to evaluate cytotoxicity of tested hybrid nanomaterials and its properties. We believe that is good starting point for further analysis and publication.

Thank you for pointing out the interesting idea for the new research and publication

  1. Section 2.3. HCPT loading and release. Needs extensive corrections. Not OK. Why not a simple release data, preferably, UV/Vis obtained?

Answer:

Thank you for this remark. The result section has been modified as was recommended by the Reviewer#1 (all corrections have been marked in red).

  1. Line 219-223. Need serious discussion and avoid stating the future goals line 222-223 (should be part of conclusion).

Answer:

Thank you for this remark. The result section has been modified as was recommended by the Reviewer#1 (all corrections have been marked in red).

  1. 269-305. A lot of literature. Kindly make the discussion to the point and avoid provision of non-specific details.

Answer:

Thank you for this issue. The section has been modified as was recommended (all corrections have been marked in red).

  1. Line 350. Cytotoxicity was evaluated on which normal cell lines? Mention that. Also, if the hybrids are toxic to normal cell lines in even in very small doses, that could render the material unsafe.

Answer:

Thank you for this remark. The cytotoxicity of tested nanomaterial was also tested on normal L929 cell line.

  1. Line 367-394. Same comment as number 11. A lot of literature.

Answer:

Thank you for this remark. This section has been modified as was recommended by the Reviewer#1 (all corrections have been marked in red).

  1. Section: 3.4. line 633. Modifications in drug structure might resulted. Did you perform any NMR studies of resulted drug-derivatives?

Answer:

We appreciate the reviewer's insightful comment. It should be noticed that the presented paper has preliminary character. Consequently, use of that kind of research gives possibility to evaluate cytotoxicity of tested hybrid nanomaterials and its properties. We believe that is good starting point for further analysis and publication.

  1. 644. Correct °C. check all the MS for such type of mistakes.

Answer:

Thank you for drawing our attention to the mistake. The manuscript has been corrected.

  1. Line 652. How are you sure that there only Fe3O4 and no Fe2O3 in the nano form?

Answer:

It is difficult to separate magnetite Fe3O4 and maghemite γ-Fe2O3 phases in XRD pattern because both have a spinel-type crystal structure. In order to verify reliably the iron oxide phase, we carried out Mössbauer spectroscopy measurements which very sensitive for iron valence state (Fe3+, Fe2+).

Figure. X-ray diffraction patterns of the Fe3O4 /Fe2O3

Moreover, the Fe3O4 nanoparticles are black (or a brownish black), while Fe2O3 nanoparticles are typically red to reddish-brown.

  1. 655-665. No need of discussing the KBr disc preparation.

Answer:

We appreciate your recommendation concerning this issue. According to Reviewer #1 suggestion the changes have been made.

  1. Line 674. Why 325 nm and 365 nm (line 684) were selected for the measurements?

Answer:

The 325 nm and 365 nm were selected as the UV–Vis specta for GO and Fe analysis.

  1. All most all the references are very old. 

Answer:

Thank you for paying attention to this fact. We have revised carefully the manuscript and references have been changed (marked in red in the main text).

Reviewer 2 Report

Comments and Suggestions for Authors

The manuscript titled “The influence of graphene oxide – Fe3O4 differently conjugated with 10-hydroxycapthotecin and rotating magnetic field on adenocarcinoma cells” by Jerdrzejczak-Silicka, M.; et al. is an original scientific work where the authors studied the impact of grafene oxide with ferrite (Fe3O4) globular nanoparticles grafted with 10-hydroxycampthotecin (HCPT) on the viability of malignant MCF-7 breast cancer cells under the exposure of rotating magnetic fields. The authors employed many complementary techniques to fully characterize the performance of the synthesized material. This could be a preliminary step to develop customized targeted-strategies not only suitable in cancer treatments, but also expandable for other human diseases.

However, it exists some points that need to be addressed (please, see them below detailed point-by-point). The most relevant outcomes remarked by the authors can contribute in the growth of many fields like the design of the next-generation of magnetic nanoparticles for drug delivery and hyperthermia treatment against cancer diseases. For this reason, I will recommend the present scientific manuscript for further publication in the International Journal of Molecular Sciences once all the below described suggestions will be properly fixed.

Here, there exists some points that must be covered in order to improve the scientific quality of the manuscript paper:

1) TITLE. “10-hydroxycapthotecin” should be modified by “10-hydroxycampthotecin”. Please, the authors need to fix this issue.

2) KEYWORDS. (OPTIONAL) The authors should consider to add the term “drug delivery strategies” in the keyword list.

3) INTRODUCTION. “According to the World Cancer Report 2014 (…) (and 8.2 million cancer-realted deaths) in 2012” (lines 36-39). Please, the authors should add more recently reported quantitative information details about the worldwide incidence of human cancer diseases [1].

[1] Kocarnik, J.M.; et al. Cancer Incidence, Mortality, Years of Life Lost, Years Lived With Disability, and Disability-Adjusted Life Years for 29 Cancer Groups From 2010 to 2019. JAMA Oncol. 2022, 8, 420-444. https://doi.org/10.1001/jamaoncol.2021.6987.

4) “A specific cancer (…) appropiate treatment (…) traditional treatment” (lines 44-45). This sentence should be rephrased in order to avoid redundancies.

5) “The nanoparticles targeted (…) determined by the magnetic properties of the NPs (…) magnetic field" (lines 65-67). Here, the authors need to introduce bulk [2] and single molecule [3] techniques to characterize the intrinsic magnetic properties of the NPs.

[2] Calvo, R.; et al. Novel Characterization Techniques for Multifunctional Plasmonic-Magnetic Nanoparticles in Biomedical Applications. Nanomaterials 2023, 13, 2929. https://doi.org/10.3390/nano13222929.

[3] Winkler, R.; et al. A Review of the Current State of Magnetic Force Microscopy to Unravel the Magnetic Properties of Nanomaterials Applied in Biological Systems and Future Direction for Quantum Technologies. Nanomaterials 2023, 13, 2585. https://doi.org/10.3390/nano13182585.

6) “Chen et al. presented (…) doxorubicin hydrochloride (DOX) –ananti-cancer drug (…) drug delivery” (lines 76-78). Please, the authors should modify the aforementioned statement by “(…) (DOX) – an anti-cancer drug (…)”.

7) RESULTS AND DISCUSSION. (OPTIONAL) The authors should consider to merge the current Figure 1 (line 123) and Figure 2 (line 124) in one single Figure. Then, the scale bars should be homogenized in order to better compare the dimensions of the visualized features among the different tested conditions (e.g. panel (a) and (b) of Fig. 1).

8) Figure 3, panels (c) and (d) (line 142). The Gaussian fitting should be plotted for each particle size distribution. Then, the number of counts (Y-axis) should be also detailed.

9) Figure 4 (line 147). Please, the authors should homogenize the range of values depicted in the X-axis of GO spectrum (~8º-50º) compared to the rest of spectra (5º-70º).

10) “The stability of the dispersions of nc-GO-Fe3O4-HCPT and c-GO- Fe3O4-HCPT nanocomposites” subsection (lines 164-178). Why did the authors not monitore longer times than 130 hours since a plateau is not reached in any of the tested conditions?

11) “Moreover, (…) R2 ranged from 0,911 to 0,9924” (line 208). Please, the authors should homogenize the significant figures. Then, the “commas” should be exchanged by “points”. These aspects should be taken into account for the rest of the main manuscript body text.

12) Figure 8 (line 320). The figure caption states “p-values<0.05 were considered significant and were represented by different small letters” but this information was not found in this Figure. Same comment for the Fig. 9 (line 395) and the Fig. 10 (line 422). The authors should indicate in the respective Figure caption that these details appear as Supplementary Information.

13) “2.8. Potential co-effect of GO-Fe3O4 nanoparticles loaded with HCPT and RMF on the viability of MCF-7 cells” (lines 427-574). Did the authors observe any nanoparticle aggregation effect? A brief statement should be provided in this regard (Some of this information was already partially discussed in the Fig.S1).

14) MATERIALS AND METHODS. Some of the sections request to be further described. Were the graphene oxide and magnetite nanospheres synthetisized under protected atmosphere (e.g. Nitrogen or Argon)?

What was the acceleration electron voltage (in keV) used in the TEM measurements?

What was the operational mode and probes selected in the AFM measurements?

15) Then, the authors should also specify the software tools employed to process the raw data obtained by all the techniques involved in this research.

16) CONCLUSIONS. This section perfectly remarks the most relevant outcomes found in this research. No actions are requested from the authors.

16) REFERENCES. The references are in the proper format style of the International Journal of Molecular Sciences once. No actions are requested from the authors.

Comments on the Quality of English Language

The submitted manuscript is generally well-written. However, it may be advisable to do a final check in order to polish those aspects susceptible to be improved. 

Author Response

Journal:                      International Journal of Molecular Science

Manuscript ID:           ijms-2776466R1

Manuscript title:        The influence of graphene oxide – Fe3O4 differently conjugated with 10-hydroxycapthotecin and rotating magnetic field on adenocarcinoma cells

Auhors:                      Magdalena Jedrzejczak-Silicka, Karolina SzymaÅ„ska, Ewa Mijowska, RafaÅ‚ Rakoczy

Reply for the Editors’ and Reviewers' comments

We would like to thank you very much for the valuable comments concerning our manuscript ” The influence of graphene oxide – Fe3O4 differently conjugated with 10-hydroxycapthotecin and rotating magnetic field on adenocarcinoma cells” (ijms-2776466) to International Journal of Molecular Sciences. We would like also to thank the Reviewers very much for their effort in the revision of the manuscript and all insightful comments and helpful suggestions which lead to a significant improvement of our manuscript. All suggestions were strong taken into consideration what surely has improved the value of the paper which reports, we believe, a very important message.

We hope that our corrections, answers and comments will be accepted by the Reviewers and the improved manuscript will be approved for publication in the International Journal of Molecular Science.

Editors and Reviewers #1 comments are in bold sans serif text.

Responses to Reviewers #1 comments are in un-bolded, serif text.

All the corrections in the manuscript have been marked in red color.

Reviewer #2

The manuscript titled “The influence of graphene oxide – Fe3O4 differently conjugated with 10-hydroxycapthotecin and rotating magnetic field on adenocarcinoma cells” by Jerdrzejczak-Silicka, M.; et al. is an original scientific work where the authors studied the impact of grafene oxide with ferrite (Fe3O4) globular nanoparticles grafted with 10-hydroxycampthotecin (HCPT) on the viability of malignant MCF-7 breast cancer cells under the exposure of rotating magnetic fields. The authors employed many complementary techniques to fully characterize the performance of the synthesized material. This could be a preliminary step to develop customized targeted-strategies not only suitable in cancer treatments, but also expandable for other human diseases.

However, it exists some points that need to be addressed (please, see them below detailed point-by-point). The most relevant outcomes remarked by the authors can contribute in the growth of many fields like the design of the next-generation of magnetic nanoparticles for drug delivery and hyperthermia treatment against cancer diseases. For this reason, I will recommend the present scientific manuscript for further publication in the International Journal of Molecular Sciences once all the below described suggestions will be properly fixed.

Here, there exists some points that must be covered in order to improve the scientific quality of the manuscript paper:

1) TITLE. “10-hydroxycapthotecin” should be modified by “10-hydroxycampthotecin”. Please, the authors need to fix this issue.

 Answer:

     We would like to thank the Reviewer for paying our attention to the mistake in the Title. The title has been corrected (marked in red).

2) KEYWORDS. (OPTIONAL) The authors should consider to add the term “drug delivery strategies” in the keyword list.

  Answer:

     In accordance with the Reviewer's request, we have added the term “drug delivery strategies” in the keyword list (marked in red).

3) INTRODUCTION. “According to the World Cancer Report 2014 (…) (and 8.2 million cancer-realted deaths) in 2012” (lines 36-39). Please, the authors should add more recently reported quantitative information details about the worldwide incidence of human cancer diseases [1].

[1] Kocarnik, J.M.; et al. Cancer Incidence, Mortality, Years of Life Lost, Years Lived With Disability, and Disability-Adjusted Life Years for 29 Cancer Groups From 2010 to 2019. JAMA Oncol. 2022, 8, 420-444. https://doi.org/10.1001/jamaoncol.2021.6987.

  Answer:

            In accordance with the Reviewer's request, the introduction has been modified (marked in red).

4) “A specific cancer (…) appropiate treatment (…) traditional treatment” (lines 44-45). This sentence should be rephrased in order to avoid redundancies.

   Answer:

Thank you for paying attention to this fact. This sentence has been corrected (marked in red in the main text).

5) “The nanoparticles targeted (…) determined by the magnetic properties of the NPs (…) magnetic field" (lines 65-67). Here, the authors need to introduce bulk [2] and single molecule [3] techniques to characterize the intrinsic magnetic properties of the NPs.

[2] Calvo, R.; et al. Novel Characterization Techniques for Multifunctional Plasmonic-Magnetic Nanoparticles in Biomedical Applications. Nanomaterials 2023, 13, 2929. https://doi.org/10.3390/nano13222929.

[3] Winkler, R.; et al. A Review of the Current State of Magnetic Force Microscopy to Unravel the Magnetic Properties of Nanomaterials Applied in Biological Systems and Future Direction for Quantum Technologies. Nanomaterials 2023, 13, 2585. https://doi.org/10.3390/nano13182585.

 Answer:

We have appreciated the recommendation by the Reviewer. The Introduction has been corrected and recommended references have been added to the text (marked in red in the main text).

6) “Chen et al. presented (…) doxorubicin hydrochloride (DOX) –ananti-cancer drug (…) drug delivery” (lines 76-78). Please, the authors should modify the aforementioned statement by “(…) (DOX) – an anti-cancer drug (…)”.

 Answer:

Thank you for paying attention to this fact. This sentence has been corrected (marked in red in the main text).

7) RESULTS AND DISCUSSION. (OPTIONAL) The authors should consider to merge the current Figure 1 (line 123) and Figure 2 (line 124) in one single Figure. Then, the scale bars should be homogenized in order to better compare the dimensions of the visualized features among the different tested conditions (e.g. panel (a) and (b) of Fig. 1).

 Answer:

Figure 1 and Figure 2 have been presented in one figure.

8) Figure 3, panels (c) and (d) (line 142). The Gaussian fitting should be plotted for each particle size distribution. Then, the number of counts (Y-axis) should be also detailed.

 Answer:

Thank you for paying attention to this fact. The Y-axis has been corrected.

9) Figure 4 (line 147). Please, the authors should homogenize the range of values depicted in the X-axis of GO spectrum (~8º-50º) compared to the rest of spectra (5º-70º).

  Answer:

Thank you for paying attention to this fact. The X-axis has been corrected.

10) “The stability of the dispersions of nc-GO-Fe3O4-HCPT and c-GO- Fe3O4-HCPT nanocomposites” subsection (lines 164-178). Why did the authors not monitore longer times than 130 hours since a plateau is not reached in any of the tested conditions?

  Answer:

            We decided to monitor the stability of the dispersions of nc-GO-Fe3O4-HCPT and c-GO- Fe3O4-HCPT nanocomposites no longer than 130 hours because the tested nanocomposites was incubated in cell culture for 48 hours and 72 hours.

11) “Moreover, (…) R2 ranged from 0,911 to 0,9924” (line 208). Please, the authors should homogenize the significant figures. Then, the “commas” should be exchanged by “points”. These aspects should be taken into account for the rest of the main manuscript body text.

  Answer:

Thank you for this remark. The result section has been modified as was recommended by the Reviewer#1 (all corrections have been marked in red).

12) Figure 8 (line 320). The figure caption states “p-values<0.05 were considered significant and were represented by different small letters” but this information was not found in this Figure. Same comment for the Fig. 9 (line 395) and the Fig. 10 (line 422). The authors should indicate in the respective Figure caption that these details appear as Supplementary Information.

 Answer:

            Thank you for paying attention to this fact. In the respective Figure caption information about p-values have been corrected as appear in Supplementary Information (marked in red in the main text).

13) “2.8. Potential co-effect of GO-Fe3O4 nanoparticles loaded with HCPT and RMF on the viability of MCF-7 cells” (lines 427-574). Did the authors observe any nanoparticle aggregation effect? A brief statement should be provided in this regard (Some of this information was already partially discussed in the Fig.S1).

  Answer:

            Thank you for paying attention to this fact. This part of the manuscript has been corrected as (marked in red in the main text).

14) MATERIALS AND METHODS. Some of the sections request to be further described. Were the graphene oxide and magnetite nanospheres synthetisized under protected atmosphere (e.g. Nitrogen or Argon)?

Answer:

            Agron has been used as a protected atmosphere.

What was the acceleration electron voltage (in keV) used in the TEM measurements?

Answer:

            The morphology of the samples was explored a high-resolution transmission electron microscopy (HRTEM) operated at 300 kV accelerating voltage.

What was the operational mode and probes selected in the AFM measurements?

 Answer:

                Atomic force microscopy (AFM) using a contact-mode AFM (CM-AFM) (Nanoscope V Multimode 8, Bruker AXS, Mannheim, Germany) has been used to investigate the morphology, thickness and size of graphene oxide flakes.

15) Then, the authors should also specify the software tools employed to process the raw data obtained by all the techniques involved in this research.

 Answer:

                Information about software tools has been added to the text (marked in red in the main text).

Round 2

Reviewer 1 Report

Comments and Suggestions for Authors

Dear Authros

I am satisfied.

Reviewer 2 Report

Comments and Suggestions for Authors

The authors did a significant deal of effort to cover all the comments raised by the Reviewers. For this reason, the scientific manuscript quality was greatly improved. Based on the significance of the topic addressed by this Review work and the scope of the International Journal of Molecular Sciences (IJMS) I would like to warmly endorse this work for further publication in this journal.